# Unlocking weather observations from the Societas Meteorologica Palatina (1781-1792)

Duncan Pappert[1,2], Yuri Brugnara[1,2], Sylvie Jourdain[3], Aleksandra Pospieszyńska[4,5], Rajmund Przybylak[4,5], Christian Rohr[2,6], and Stefan Brönnimann[1,2]

[1]Institute of Geography, University of Bern, Bern, Switzerland
[2]Oeschger Centre for Climate Research, University of Bern, Bern, Switzerland
[3]Direction de la Climatologie et des Service Climatiques, Météo-France, Toulouse, France
[4]Faculty of Earth Sciences and Spatial Management, Nicolaus Copernicus University, Toruń, Poland
[5]Centre for Climate Change Research, Nicolaus Copernicus University, Toruń, Poland
[6]Institute of History, University of Bern, Bern, Switzerland

**Correspondence:** Duncan Pappert (duncan.pappert@giub.unibe.ch); Yuri Brugnara (yuri.brugnara@giub.unibe.ch)

**Abstract.** Recent years have seen early instrumental observations play an increasingly important role in climate research, allowing past daily-to-decadal climate variability and weather extremes to be explored in greater detail. The 18th century saw the formation of several short-lived meteorological networks of which the one organised by the *Societas Meteorologica Palatina* is arguably the most well-known. This network stood out as one of those few that efficiently managed to control its members, integrating, refining and publishing measurements taken from numerous stations around Europe and beyond. Although much has been written about the network in both history, science, and individual prominent series used for climatological studies, the actual measurements have not yet been digitised and published in extenso. This paper represents an important step towards filling this perceived gap in research. Here we provide an inventory listing the availability of observed variables for the 37 stations that belonged to the Society's network and discuss their historical context. Most of these observations have been digitised and a considerable fraction has been converted and formatted. In this paper we focus on the temperature and pressure measurements, which have been corrected and homogenised. We then demonstrate their potential for climate research by analysing two cases of extreme weather. The recovered series will have wide applications and could contribute to a better understanding of the mechanisms behind climatic variations and extremes, as well as the societal reactions to adverse weather. Even the shorter series could be ingested into reanalyses and improve the quality of large scale reconstructions.

## 1 Introduction

For some decades now, early instrumental observations have increasingly played an important role in climate research (see Jones, 2001; Brönnimann et al., 2019). Not only have they shown enormous potential in reconstructing past changes in temperature, pressure, and precipitation, but they also allow for daily-to-decadal variability and extremes to be studied in great detail, particularly in 18th- and early 19th-century Europe (e.g. Brázdil et al., 2010; Camuffo et al., 2010; Csernus-Molnár et al., 2014; Pfister et al., 2019, 2020). After the 1770s, meteorologists were making substantial contributions to the design of precise instruments of physics. More and more members of the scientific community were calling for the improved quantification and

standardisation of weather observations, as well as a more rigorous discipline in recording this information in a consistent manner that would allow for a cohesive characterisation of climate. The late 18th century witnessed a greater synthetic vision among meteorologists, marking a departure from the chaotic data gathering style that had characterised the preceding decades (Feldman, 1983).

In the late 18th century, a number of short-lived national and international networks were zealously pursuing a more organised form of meteorology. In 1776, two years after publishing his influential *Treaty on Meteorology*, Père Louis Cotte began compiling data from several locations in France for the Société royale de médecine, based in Paris (Cotte, 1774). In Padua, Director of the Astronomical Observatory Giuseppe Toaldo, whose prize-winning work on the physical influence of weather on agriculture would spurn the optimism of other networks (Toaldo, 1775), had been taking meteorological measurements since 1766. Additionally, Toaldo set up a network of precipitation measurements in Italy during the 1780s that functioned for nearly 30 years. Another meteorological programme was organised by the Oekonomische Gesellschaft Bern, which had distributed instruments to a number of stations in the then Republic of Bern (OeGB, 1762; Pfister, 1975). Other networks included the early Dutch meteorological society, Natuur- en Geneeskundige Correspondentie Sociëteit (1779–1802), led by Jan Hendrik van Swinden, and the Spanish network, Real Academia de Médica Matritense (1780–1825), based in Madrid.

The *Societas Meteorologica Palatina*'s meteorological network (henceforth referred to as SMP), or Palatine Meteorological Society, is commonly regarded as the most successful and prolific among these organisations, whose work, according to Theodore S. Feldman, "was not surpassed for three-quarters of a century" (Feldman, 1990, p. 154). Founded in 1780 by the Elector Palatine of Bavaria Karl Theodor and directed by his secretary Johannes Jakob Hemmer, the SMP stood out as one of those few networks that efficiently managed to control its members, integrating, refining and publishing measurements taken from numerous stations around Europe and beyond (Kington, 1974; Cassidy, 1985). Eventually, the Society's increasing financial difficulties together with the political confusion and social turmoil caused by the French Revolution brought about the collapse of the Society in 1795 – which had already been in decline after the death of Hemmer in 1790 – and brought a swift decline to meteorology; this slump would last until the start of continuous measurements by national networks in the 1830s. Nevertheless, what the SMP and its members achieved in the late 18th century in terms of coordination and discipline of observations meant that researchers today have a wealth of more detailed and organised information about the weather of the time.

Despite enjoying relatively great fame among historical climatologists, the abundant weather information in the SMP's publications has yet to be understood in more depth. A lot of research has followed a historical approach, focusing on the origins and development of the SMP and its members (Traumüller, 1885; Feldman, 1983; Cassidy, 1985; Wege and Winkler, 2004; Lüdecke, 2004, 2010; Vinther et al., 2006). Data from the Society have been used for climatological analysis, though mostly making use of single stations (Przybylak et al., 2014; Raicich et al., 2015; Camuffo et al., 2017; Häderli et al., 2020), rarely looking at more than a couple (Brázdil et al., 2010; Yiou et al., 2014). Aspaas and Hansen (2012) analysed occurrences of the aurora borealis recorded by the different stations as well as the phenological comments in the SMP's annual reports. In his book *The Weather of the 1780s over Europe*, Kington (1988) used the available station data from this decade – including several stations from the SMP – to create hand-drawn daily synoptic weather maps, showing that prevailing weather situations

Climate of the Past Discussions Open Access

over Europe at the time (with regard to moving pressure systems) could be reconstructed and analysed. More recently, various data rescue projects such as IMPROVE (Camuffo and Jones, 2002), HISTALP (Auer et al., 2007), the ANR project CHEdaR (2010-2014), and CHIMES (Brugnara et al., 2020), have incorporated larger portions of the data for reconstructing long high-quality series. Despite these important efforts, only a small fraction of the overall measurements have been used in modern climate research and the data had yet to be digitised or published in extenso. There is much more to be done with regard to unlocking the full potential of the data gathered by the SMP over 230 years ago.

This paper makes an important step toward filling this perceived gap in research. In a first part, this study provides an inventory – attached as .csv file in the supplement – listing exactly what kind of weather observations each station reported to the SMP. In other words, one of the aims is to create an inventory containing significant metadata, making tracking and working with them easier in the future. The next objective is to convert the digitised data into modern units and make them ready for scientific research. The focus of this paper is on the temperature and pressure measurements of the network.

What does the data rescue reveal about the availability and quality of the meteorological observations in the SMP's publications? What is the potential of its temperature and pressure data for climate research? To better explore these questions this paper is organised as follows. In Sect. 2 we provide a detailed explanation of the data rescue procedure and the corrections applied to the temperature and pressure observations, as well as discuss some of the issues that may have affected the measurements. In Sect. 3 we present and discuss the main findings of the inventory. We then demonstrate the usefulness of these series by examining two examples in Sect. 4, after which we make our concluding remarks.

## 2 Data and Methods

### 2.1 Source material description

The materials used in this study are mainly the *Ephemerides*, the yearly publications by the *Societas Meteorologica Palatina*, containing the collated measurements from the network. These are twelve volumes published from 1783 to 1795, containing observations for the period 1781-92. The original volumes have already been scanned and can be found – among other places – online on the website of the Ludwig Maximilian University of Munich Library.

In the period 1781-92 a total of 37 stations contributed to Hemmer's project (see Table 1). The stations covered a wide area of mainland Europe, stretching from Rome in the South, La Rochelle in the West, and reaching as far North-East as St Petersburg and Moscow (Fig. 1). As such, the data of the SMP are well-placed for an extensive study to be made of Europe's climate variability and extreme events during that decade. Notably, there was a number of stations beyond mainland Europe that was also part of the SMP, which included Cambridge, Massachusetts in North America, Godthaab in Greenland, and Pyshminsk in Siberia.

Table 1: List of stations that belonged to the SMP network, including information about height above sea level (estimations based on available information), longitude, latitude, and period for which there is data in the *Ephemerides*. This is loosely based on a list by Cassidy (1985, pp. 23-4), which does not specify the resolution of the observations.

| Station | Abbr. | z (m a.s.l.) | $\phi$ (°) | $\lambda$ (°) | Period covered |
|---|---|---|---|---|---|
| Andechs | AND | 710 | 47.97 | 11.18 | 1781-92 |
| Berlin | BRL | 45 | 52.52 | 13.40 | 1781-88 |
| Bologna | BOL | 74 | 44.49 | 11.34 | 1782-84, 87-92 |
| Brussels | BRU | 60 | 50.85 | 4.35 | 1782-92 |
| Budapest | BUD | 168 | 47.50 | 19.04 | 1781-92 |
| Cambridge (US) | CAM | 9 | 42.37 | -71.11 | 1782-87 |
| Copenhagen | COP | 40 | 55.68 | 12.57 | 1782-88 |
| Delft | DEL | 2 | 52.10 | 5.19 | 1784, 86 |
| Den Haag | HAG | 2 | 52.07 | 4.30 | 1782-83 |
| Dijon | DIJ | 250 | 47.32 | 5.04 | 1783-84 |
| Düsseldorf | DUS | 30 | 51.23 | 6.77 | 1782-84 |
| Eidsberg | EDS | 70 | 59.54 | 11.36 | 1787 |
| Erfurt | ERF | 202 | 50.98 | 11.03 | 1781-88 |
| Geneva | GEN | 380 | 46.20 | 6.14 | 1782-89 |
| Godthaab (GL) | GDH | – | 64.17 | -51.75 | 1786-87 |
| Gotthard | GOT | 2093 | 46.55 | 8.57 | 1781-92 |
| Göttingen | GTN | 150 | 51.53 | 9.93 | 1783-85, 87 |
| Hohenpeissenberg | HOH | 977 | 47.80 | 11.01 | 1781-92 |
| Ingolstadt | ING | 369 | 48.75 | 11.43 | 1781-82 |
| La Rochelle | ROC | 19 | 46.16 | -1.15 | 1782-90 |
| Mannheim | MAN | 112 | 49.49 | 8.47 | 1781-92 |
| Marseille | MAR | 44 | 43.30 | 5.36 | 1782-92 |
| Middelburg | MID | 15 | 51.49 | 3.61 | 1782-88 |
| Moscow | MOS | 130 | 55.76 | 37.62 | 1783-89, 91-92 |
| Munich | MUN | 525 | 48.13 | 11.58 | 1781-92 |
| Padua | PAD | 18 | 45.41 | 11.88 | 1781-92 |
| Prague | PRA | 200 | 50.07 | 14.44 | 1781-87, 89-91 |
| Pyshminsk (RU) | PYS | – | 56.98 | 60.59 | 1789-90 |

| | | | | | |
|---|---|---|---|---|---|
| Regensburg | REG | 346 | 49.01 | 12.10 | 1781-91 |
| Rome | ROM | 56 | 41.90 | 12.50 | 1782-92 |
| Spydeberg | SPY | 110 | 59.62 | 11.08 | 1783-86 |
| St Petersburg | PET | 15 | 59.93 | 30.36 | 1783-92 |
| St Zeno | ZEN | 480 | 47.73 | 12.88 | 1781 |
| Stockholm | STO | 44.4 | 59.34 | 18.06 | 1783-92 |
| Tegernsee | TEG | 740 | 47.72 | 11.77 | 1781-89 |
| Würzburg | WUR | 200 | 49.79 | 9.93 | 1781-88 |
| Żagań | ZAG | 116 | 51.37 | 15.19 | 1781-92 |

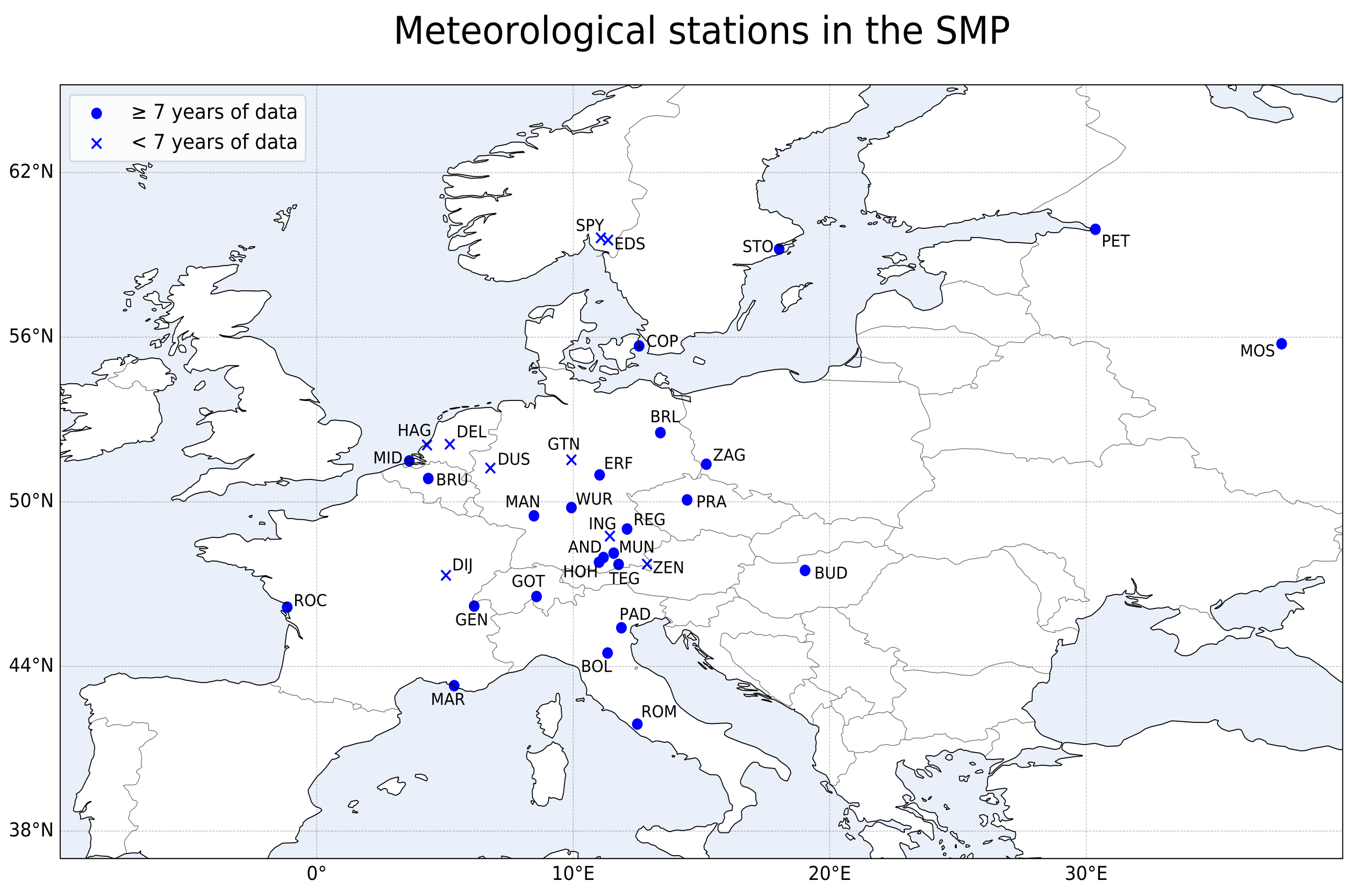

**Figure 1.** Network of stations that contributed to the *Ephemerides* of the *Societas Meteorologica Palatina* – see Table 1 for abbreviations. Dots represent those stations that contributed for 7 or more years of the total 12-year period 1781-92; locations with a cross are stations that sent measurements for fewer than 7 years. This distinction is made to broadly convey the varying extent of the stations' participation. Not shown on the map are the stations at Cambridge (US), Godthaab (GL), and Pyshminsk (RU).

In 1780, working in Mannheim, Carlo Artaria began to manufacture the meteorological instruments that were to be distributed to the measuring sites of the network. These instruments were a mercury barometer, two mercury thermometers, a hygrometer, and for some selected sites, a magnetic declinometer (Grenon, 2010), all of which had been standardised, gauged, and calibrated to be almost identical, so that comparing measurements at a later time would be more straightforward. Other instruments, such as the pluviometer, wind vane, etc. were to be constructed locally by the observers; these instruments did not require any calibration and only needed to be placed according to Hemmer's instructions. The instruments are described in great detail in their first publication (*Ephemerides*, 1783, pp. 59-90). Using scientific Latin to overcome language differences, the SMP specified how to work with the instruments, where to place them, when to take the measurements, and in what style the observations should be written down. They were instructed to take measurements three times a day, ideally at 07:00, 14:00, and 21:00 UTC+1, of pressure, temperature, precipitation, magnetic declination, wind direction and force, evaporation, height of rivers and lakes, clouds and state of the sky, using common units and symbology. The data was then dispatched to the headquarters in Mannheim, where it was assembled and published in the annual *Ephemerides Societatis Meteorologicae*.

The *Ephemerides* are structured in two parts; the main portion being the *observationes integrae* (extended observations), containing the subdaily measurements and observations, and the *appendix* or *compendium redactae* (synopsis), which comprises the monthly means and other summary statistics. The publications for 1781 and 1792 only contain the extended observations. For 1787-88 the format was temporarily changed to comprise a *pars prior* and a *pars posterior*, respectively containing a smaller subdaily section and an extended appendix with both daily means and monthly summaries (see Sect. 3). In some volumes, there is an *additamentum* (addition) to include observations that had not made it to print for the previous publications.

Figure 2 shows an example for the typical arrangement of subdaily data in the publications. The table is clear and straightforward: above the table are written the station, the name of the observer, the observation times, and the month in question; the table itself contains for each day measurements from the barometer (pressure), indoor thermometer (room temperature), outdoor thermometer (air shade temperature), hygrometer (humidity), declinometer (magnetic declination), wind vane (wind direction and force), rain gauge (rainfall), and an atmidometer (evaporation). River or lake height measurements were also taken where possible (in the example of Budapest in Figure 2, for the river Danube). Additionally, the SMP adopted a common symbology to describe the phase of the moon, the state of the sky and significant weather and special phenomena, including rain, hail, snow, frost, fog and thunderstorms. An illustration of these symbols can be found in Kington (1988, p. 24).

### 2.2 Compilation of measurements

Despite the coherent overarching concept guiding collection of data in the *Ephemerides*, the content of the publications is more variegated than at first meets the eye. To enable a more systematic understanding of the weather information these publications provide to historical climatologists, this study presents the meteorological observations in a way that is easier to visualise and analyse. For each station of the network, the inventory lists for which years they reported measurements, what type of resolution (subdaily, daily, monthly), which variables (temperature, pressure, rainfall, etc.), and the name of the observers. This inventory can be maintained as a living document and updated by future researchers who study the *Ephemerides* and represents an important step in the systematic rescue and evaluation of the SMP's observations; it can be found as a .csv file in

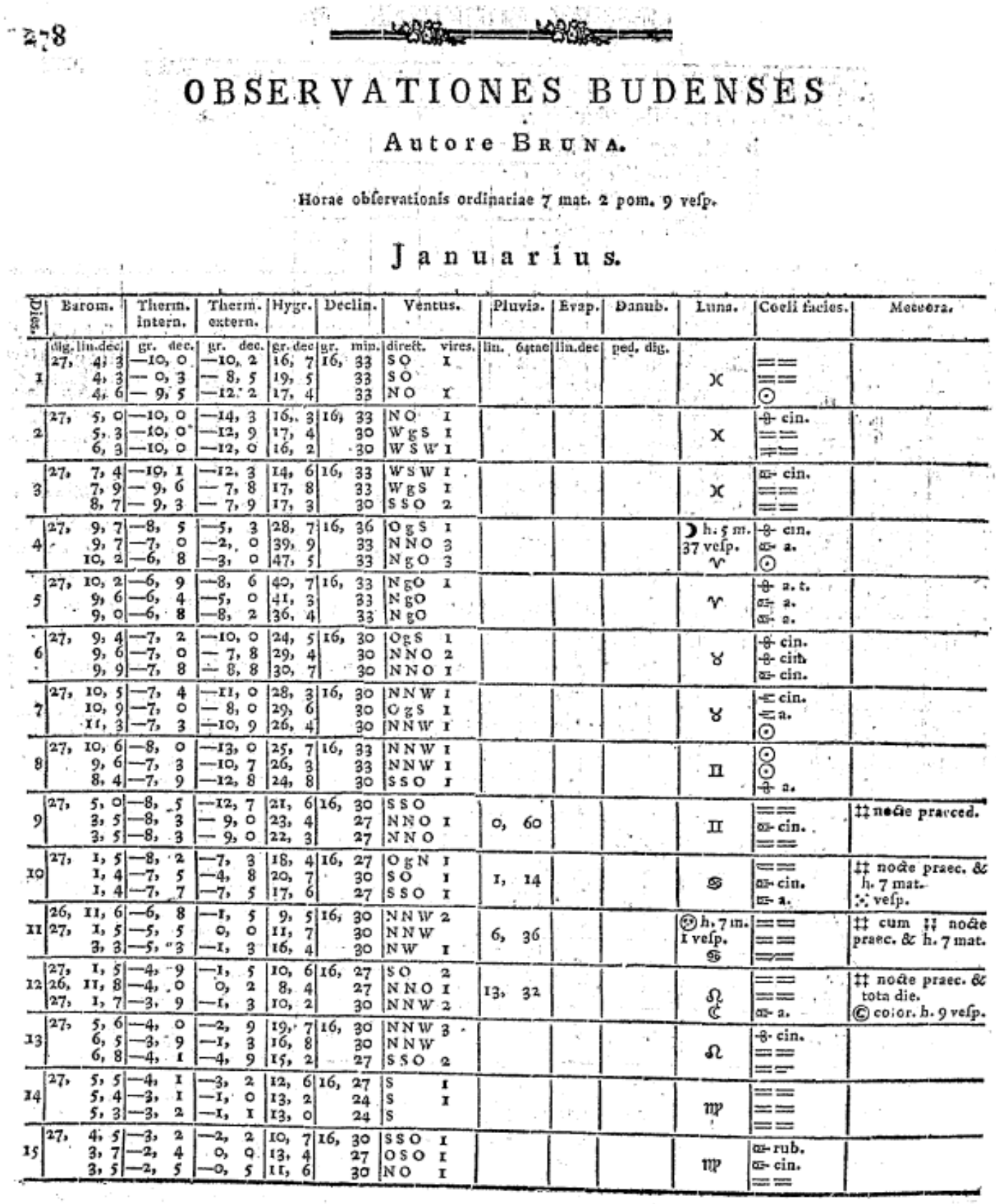

278

OBSERVATIONES BUDENSES

Autore Bruna.

Horae obſervationis ordinariae 7 mat. 2 pom. 9 veſp.

Januarius.

| Dies. | Barom. | Therm. intern. | Therm. extern. | Hygr. | Declin. | Ventus. | Pluvia. | Evap. | Danub. | Luna. | Coeli facies. | Meteora. |
|---|---|---|---|---|---|---|---|---|---|---|---|---|
| | dig. lin. dec. | gr. dec. | gr. dec. | gr. dec. | gr. min. | direct. vires. | lin. 64tae | lin. dec. | ped. dig. | | | |
| 1 | 27, 4, 3<br>4, 3<br>4, 6 | —10, 0<br>— 0, 3<br>— 9, 5 | —10, 2<br>— 8, 5<br>—12, 2 | 16, 7<br>19, 5<br>17, 4 | 16, 33<br>33<br>33 | SO 1<br>SO<br>NO 1 | | | | ♓ | ==<br>==<br>⊙ | |
| 2 | 27, 5, 0<br>5, 3<br>6, 3 | —10, 0<br>—10, 0<br>—10, 0 | —14, 3<br>—12, 9<br>—12, 0 | 16, 3<br>17, 4<br>16, 2 | 16, 33<br>30<br>30 | NO 1<br>WgS 1<br>WSW 1 | | | | ♓ | -ϑ- cin.<br>==<br>== | |
| 3 | 27, 7, 4<br>7, 9<br>8, 7 | —10, 1<br>— 9, 6<br>— 9, 3 | —12, 3<br>— 7, 8<br>— 7, 9 | 14, 6<br>17, 8<br>17, 3 | 16, 33<br>33<br>30 | WSW 1<br>WgS 1<br>SSO 2 | | | | ♓ | ⚌- cin.<br>==<br>== | |
| 4 | 27, 9, 7<br>9, 7<br>10, 2 | —8, 5<br>—7, 0<br>—6, 8 | —5, 3<br>—2, 0<br>—3, 0 | 28, 7<br>39, 9<br>47, 5 | 16, 36<br>33<br>33 | OgS 1<br>NNO 3<br>NgO 3 | | | | ☽ h. 5 m. 37 veſp.<br>♈ | -ϑ- cin.<br>⚌- a.<br>⊙ | |
| 5 | 27, 10, 2<br>9, 6<br>9, 0 | —6, 9<br>—6, 4<br>—6, 8 | —8, 6<br>—5, 0<br>—8, 2 | 40, 7<br>41, 3<br>36, 4 | 16, 33<br>33<br>33 | NgO 1<br>NgO<br>NgO | | | | ♈ | -ϑ- a. t.<br>⚌- a.<br>⚌- a. | |
| 6 | 27, 9, 4<br>9, 6<br>9, 9 | —7, 2<br>—7, 0<br>—7, 8 | —10, 0<br>— 7, 8<br>— 8, 8 | 24, 5<br>29, 4<br>30, 7 | 16, 30<br>30<br>30 | OgS 1<br>NNO 2<br>NNO 1 | | | | ♉ | -ϑ- cin.<br>-ϑ- cin.<br>⚌- cin. | |
| 7 | 27, 10, 5<br>10, 9<br>11, 3 | —7, 4<br>—7, 0<br>—7, 3 | —11, 0<br>— 8, 0<br>—10, 9 | 28, 3<br>29, 6<br>26, 4 | 16, 30<br>30<br>30 | NNW 1<br>OgS 1<br>NNW 1 | | | | ♉ | -⚌ cin.<br>-⚌ a.<br>⊙ | |
| 8 | 27, 10, 6<br>9, 6<br>8, 4 | —8, 0<br>—7, 3<br>—7, 9 | —13, 0<br>—10, 7<br>—12, 8 | 25, 7<br>26, 3<br>24, 8 | 16, 33<br>33<br>30 | NNW 1<br>NNW 1<br>SSO 1 | | | | ♊ | ⊙<br>⊙<br>-ϑ- a. | |
| 9 | 27, 5, 0<br>3, 5<br>3, 5 | —8, 5<br>—8, 3<br>—8, 3 | —12, 7<br>— 9, 0<br>— 9, 0 | 21, 6<br>23, 4<br>22, 3 | 16, 30<br>27<br>27 | SSO<br>NNO 1<br>NNO | 0, 60 | | | ♊ | ==<br>⚌- cin.<br>== | ⁑ nocte praeced. |
| 10 | 27, 1, 5<br>1, 4<br>1, 4 | —8, 2<br>—7, 5<br>—7, 7 | —7, 3<br>—4, 8<br>—7, 5 | 18, 4<br>20, 7<br>17, 6 | 16, 27<br>30<br>27 | OgN 1<br>SO 1<br>SSO 1 | 1, 14 | | | ♋ | ==<br>⚌- cin.<br>⚌- a. | ⁑ nocte praec. & h. 7 mat.<br>⁖ veſp. |
| 11 | 26, 11, 6<br>27, 1, 5<br>3, 3 | —6, 8<br>—5, 5<br>—5, 3 | —1, 5<br>0, 0<br>—1, 3 | 9, 5<br>11, 7<br>16, 4 | 16, 30<br>30<br>30 | NNW 2<br>NNW<br>NW 1 | 6, 36 | | | ⊕ h. 7 m. 1 veſp.<br>♋ | ==<br>==<br>== | ⁑ cum ⁑ nocte praec. & h. 7 mat. |
| 12 | 27, 1, 5<br>26, 11, 8<br>27, 1, 7 | —4, 9<br>—4, 0<br>—3, 9 | —1, 5<br>0, 2<br>—1, 3 | 10, 6<br>8, 4<br>10, 2 | 16, 27<br>27<br>30 | SO 2<br>NNO 1<br>NNW 2 | 13, 32 | | | ♌<br>☾ | ==<br>==<br>⚌- a. | ⁑ nocte praec. & tota die.<br>© color. h. 9 veſp. |
| 13 | 27, 5, 6<br>6, 5<br>6, 8 | —4, 0<br>—3, 9<br>—4, 1 | —2, 9<br>—1, 3<br>—4, 9 | 19, 7<br>16, 8<br>15, 2 | 16, 30<br>30<br>27 | NNW 3<br>NNW<br>SSO 2 | | | | ♌ | -ϑ- cin.<br>==<br>== | |
| 14 | 27, 5, 5<br>5, 4<br>5, 3 | —4, 1<br>—3, 1<br>—3, 2 | —3, 2<br>—1, 0<br>—1, 1 | 12, 6<br>13, 2<br>13, 0 | 16, 27<br>24<br>24 | S 1<br>S 1<br>S | | | | ♍ | ==<br>==<br>== | |
| 15 | 27, 4, 5<br>3, 7<br>3, 5 | —3, 2<br>—2, 4<br>—2, 5 | —2, 2<br>0, 0<br>—0, 5 | 10, 7<br>13, 4<br>11, 6 | 16, 30<br>27<br>30 | SSO 1<br>OSO 1<br>NO 1 | | | | ♍ | ⚌- rub.<br>⚌- cin.<br>== | |

**Figure 2.** Example of a typical table of subdaily measurements for the period 1-15 January 1789, in this case showing observations for Budapest, recorded by Ferenc Xavér Bruna (*Ephemerides*, 1791, p. 278).

the supplement to this paper.

It should be noted that we do not analyse the specific surroundings of the sites of the observatories themselves, like has been done in some studies for Hohenpeissenberg (Wege and Winkler, 2004; Winkler, 2009) and Żagań (Przybylak et al., 2014), but we rely instead to a certain degree on the adherence of the stations to the recommendations given by the SMP on the positioning of the instruments.

### 2.3 Data processing

The pressure readings were taken from a cistern barometer, filled with mercury, graduated in Paris inches and provided with a Vernier scale. The thermometers were of the mercury-in-glass type, in degrees Réaumur: one was to be mounted indoors to be affixed to the barometer to aid in the correction of the pressure readings, and the second to be mounted outside, facing north, sheltered from direct sunshine, but allowing the free passage of air. Temperature and pressure are the variables of main interest

for the data rescue and climatological analysis in this study.

The first step in the analysis of historical data that is not available electronically is the digitisation, which was done by key entry. Subsequently, the digitised data is processed: before being used for scientific purposes, the raw instrument readings need to undergo specific corrections and reductions in order to make them comparable. This includes converting the old observations into today's standard units; for temperature observations the conversion from Réaumur to Celsius is straightforward:

$$T_C = T_R \times 1.25 \quad (1)$$

The SMP's Russian stations at times used thermometers in the Delisle scale, and the station in Cambridge (US) initially sent observations taken with a Fahrenheit thermometer. For these units, the conversion to Celsius is done as follows:

$$T_C = 100 - T_D \times 2/3 \quad (2)$$

$$T_C = (T_F - 32)/1.8 \quad (3)$$

In a second step, temperature values were corrected for a positive freezing-point bias due to the tendency of the freezing point of thermometers to gradually rise (Winkler, 2009). One postulated explanation for this phenomenon was the rapid cooling of the glass walls of the thermometer after the manufacturing process; this would cause a re-alignment of the glass molecules over several years as it was being slowly compressed by external air pressure (Bellani, 1808; Yelin, 1824). The rise of the 0°C point can be corrected easily by assuming the same correction Winkler (2009) used can be applied to all stations, that is, a stepwise -0.1°C/year extended over the first six years of observations. This correction was applied to all series with the exception of Cambridge (US), Dijon, Middelburg, Moscow, Stockholm, and St Petersburg, which are known to have used instruments of their own, as well as the few stations that began sending data after 1785.

For pressure measured with a mercury barometer, the full conversion requires more steps. The first is a temperature correction to take into account the expansion of the mercury:

$$L_0 = (1 - \gamma T)L_{mm} \quad (4)$$

where $\gamma$ is the thermal expansion coefficient of mercury at 0°C ($1.82 \times 10^{-4} K^{-1}$). $T$ is the temperature of the barometer (or room temperature) in °C, $L_{mm}$ is the original observation in millimeters – for Paris inches the conversion factor is 27.07mm. It should be noted that some observers at the time would reduce their observations to other temperatures, 10°R being a common standard (Brugnara et al., 2015); this is assumed to be the case for pressure values sent to Mannheim by the Russian stations. For the daily means presented in the appendices of each *Ephemerides* volume, no barometer temperatures are given. Differences with corrected pressure series of neighbouring stations show a seasonal cycle – one that reflects the cycle of the (room) temperature of the barometer – indicating that these averages in the appendices were likely not reduced to 0°C. Thus, in order to get a good estimate of the room temperature climatology corresponding to the daily mean pressure in the tables, for each station these values were extrapolated from outdoor temperature using a linear regression based on the strong linear relationship

between indoor and outdoor temperature in the available subdaily observations of other years.

The next step is a unit conversion of the mercury column from millimetres to hectopascal (hPa) and the application of a gravity correction that takes into account the latitudinal and height-dependent variation of gravity acceleration:

$$P_0 = \rho g_{\varphi,h} L_0 \times 10^{-5} \quad (5)$$

where $P_0$ is the absolute pressure in hPa, $\rho$ is the density of mercury at 0°C ($1.35951 \times 10^4 kg\,m^{-3}$), $g_{\varphi,h}$ is the local gravity acceleration (estimated from latitude $\varphi$ and altitude $h$, see Brugnara et al. (2015)), and $L_0$ is the temperature-corrected barometric reading in mm calculated in Eq. 4. Finally, a reduction of the measurements to mean sea level is performed for use in the synoptic analysis:

$$P_{SLP} = P_0 \times exp\left(\frac{\frac{g_{\varphi,h}}{R} \times h}{T_{ext} + a \times \frac{h}{2}}\right) \quad (6)$$

where $R$ is the gas constant for dry air, $a$ is the standard lapse rate of the fictitious air column below the station, and $T_{ext}$ is the outside temperature of the station in question.

### 2.4 Data format

The Copernicus Climate Change Service (C3S) (Brönnimann et al., 2018) provides a simple but standard format for the distribution of data, called the Station Exchange Format (SEF). SEF files have a .tsv (tab-separated values) extension and follow a straightforward structure, listing basic metadata regarding the station and the series in a header. Each line of the data table represents one observation time for which are specified: year, month, day, hour, minute, period (time period of observations), value, and meta (containing any metadata that is of interest to that value). The common format is designed to allow people rescuing observations to present them for widespread use in an uncomplicated format and recognisable by software. The final aim is for such SEF files to be easily ingested into global repositories (Brunet et al., 2020).

### 2.5 Quality control

Overall, the quality of the temperature and pressure series recovered in this study is relatively high – due in no small part to the standardised thermometers and barometers made available by the SMP. That said, the presence of errors is unavoidable. To aid in their detection, the C3S Quality Control (QC) software provides R functions that identify climatic outliers (values outside the interquantile range), occurrences of equal consecutive values, and records where the observations exceed the WMO suggested tolerances for temperature and pressure time consistencies, among other quality tests (see https://cran.r-project.org/package=dataresqc). Visual plots were also used to inspect the data as an alternative test for erroneous values. Suspect values are flagged in the metadata column of the SEF files, stating the specific type of QC test that was failed.

The most common type of error reported for subdaily values by the QC function was the *wmo_time_consistency*, which flags observations that exceed the WMO-suggested tolerances for the temperatures and pressure tendency as a function of time. The Munich subdaily temperature series, covering the years 1781-86, 88, 90-92, had as many as 82 of its observations flagged with

this error, most of them in the last three years of observation. This hints at the possibility that for some periods at a time the thermometer was partially exposed to sunlight and not actually measuring air shade temperature. This lead to several afternoon observations being overestimated, thereby creating unrealistic jumps in the measured temperature diurnal cycle.

On the whole, however, the *wmo_time_consistency* flags in a given series are relatively few. Some of these errors may have been due to misprints in the publications, or they were probably misread by compilers in Mannheim; they could even have been a writing mistake by the observer himself. Figure 3 illustrates this example with Żagań pressure readings. The highlighted barometer lines are erroneous and not in line with the expected pressure tendency for a 6- or 7.5-hour time interval. One could hypothesise that the first value '0.2' was meant as '10.2', but the real measured value cannot be known and so should ideally be left as a flagged value. That said, the correction of a suspected printing error is at the discretion of the researcher and, whenever applied, a note should be made in the digitised table.

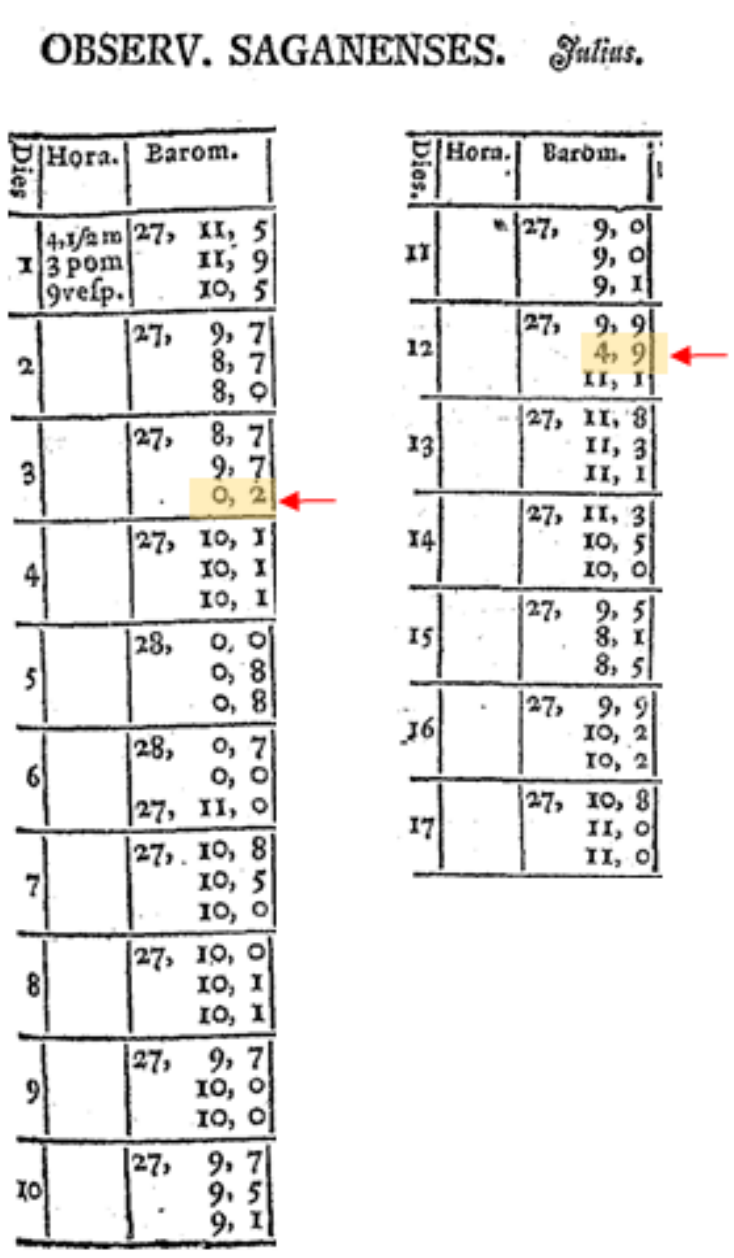
OBSERV. SAGANENSES. Julius.

| Dies | Hora. | Barom. |
|---|---|---|
| 1 | 4,1/2 m | 27, 11, 5 |
| | 3 pom | 11, 9 |
| | 9 vefp. | 10, 5 |
| 2 | | 27, 9, 7 |
| | | 8, 7 |
| | | 8, 0 |
| 3 | | 27, 8, 7 |
| | | 9, 7 |
| | | 0, 2 |
| 4 | | 27, 10, 1 |
| | | 10, 1 |
| | | 10, 1 |
| 5 | | 28, 0, 0 |
| | | 0, 8 |
| | | 0, 8 |
| 6 | | 28, 0, 7 |
| | | 0, 0 |
| | | 27, 11, 0 |
| 7 | | 27, 10, 8 |
| | | 10, 5 |
| | | 10, 0 |
| 8 | | 27, 10, 0 |
| | | 10, 1 |
| | | 10, 1 |
| 9 | | 27, 9, 7 |
| | | 10, 0 |
| | | 10, 0 |
| 10 | | 27, 9, 7 |
| | | 9, 5 |
| | | 9, 1 |

| Dies. | Hora. | Barom. |
|---|---|---|
| 11 | | 27, 9, 0 |
| | | 9, 0 |
| | | 9, 1 |
| 12 | | 27, 9, 9 |
| | | 4, 9 |
| | | 11, 1 |
| 13 | | 27, 11, 8 |
| | | 11, 3 |
| | | 11, 1 |
| 14 | | 27, 11, 3 |
| | | 10, 5 |
| | | 10, 0 |
| 15 | | 27, 9, 5 |
| | | 8, 1 |
| | | 8, 5 |
| 16 | | 27, 9, 9 |
| | | 10, 2 |
| | | 10, 2 |
| 17 | | 27, 10, 8 |
| | | 11, 0 |
| | | 11, 0 |

**Figure 3.** Barometer readings at Żagań for 1-17 July 1781 (*Ephemerides*, 1783, pp. 373-4). The columns represent the days, observations times, and Parish inches and lines. The highlighted values are likely printing errors and flagged in the SEF file as *wmo_time_consistency* errors.

A number of winter values in the temperature series were flagged as *subdaily_out_of_range* for exceeding the default threshold (-30°C) of the QC function. These were mostly ignored as they seemed reasonable given the internal variability of temperature during the 1780s. Indeed, for extreme meteorological conditions, it could be that values are flagged as suspect, though being correct (WMO, 1993).

A closer look at the Brussels temperature series revealed suspicious values in December 1788. It would seem that daily

hygrometer observations for this month were mistakenly copied into the temperature column (*Ephemerides*, 1790, *pars posterior*, pp. 38-9). The error became apparent when comparing temperature values with those of Middelburg, located just over 100km from Brussels. The QC software did not recognise this inconsistency due to the hygrometer measurements resembling a plausible range of temperature values.

On the lower end of the quality 'spectrum' of the recovered series is the Moscow pressure series (Fig. 4). The availability of data is patchy to begin with: observations started being taken in 1783 until 1792, except for 1790; 1783-84 and 1787 are available in the form of monthly summaries; the rest are in subdaily form. About 13% of the total available subdaily values were flagged as *subdaily_repetition*s, i.e. occurrences of equal consecutive values. This hints at the poor quality of the barometer that was being used. For the Moscow station, 1785 marks both a change in observer, from Engel to Stritter, and a change from noting observations in English to Paris inches. It is unclear whether there was a change in instrument: it may not have been a SMP barometer, as lines were still being reported in 1/100 instead of 1/12; on the other hand, it could be that readings from the old English barometer were simply being converted to Paris inches in the editing process. Furthermore, as can be seen in the time series, the pressure series likely needs to be homogenised, a task which could not be carried out, given the paucity of nearby reference series.

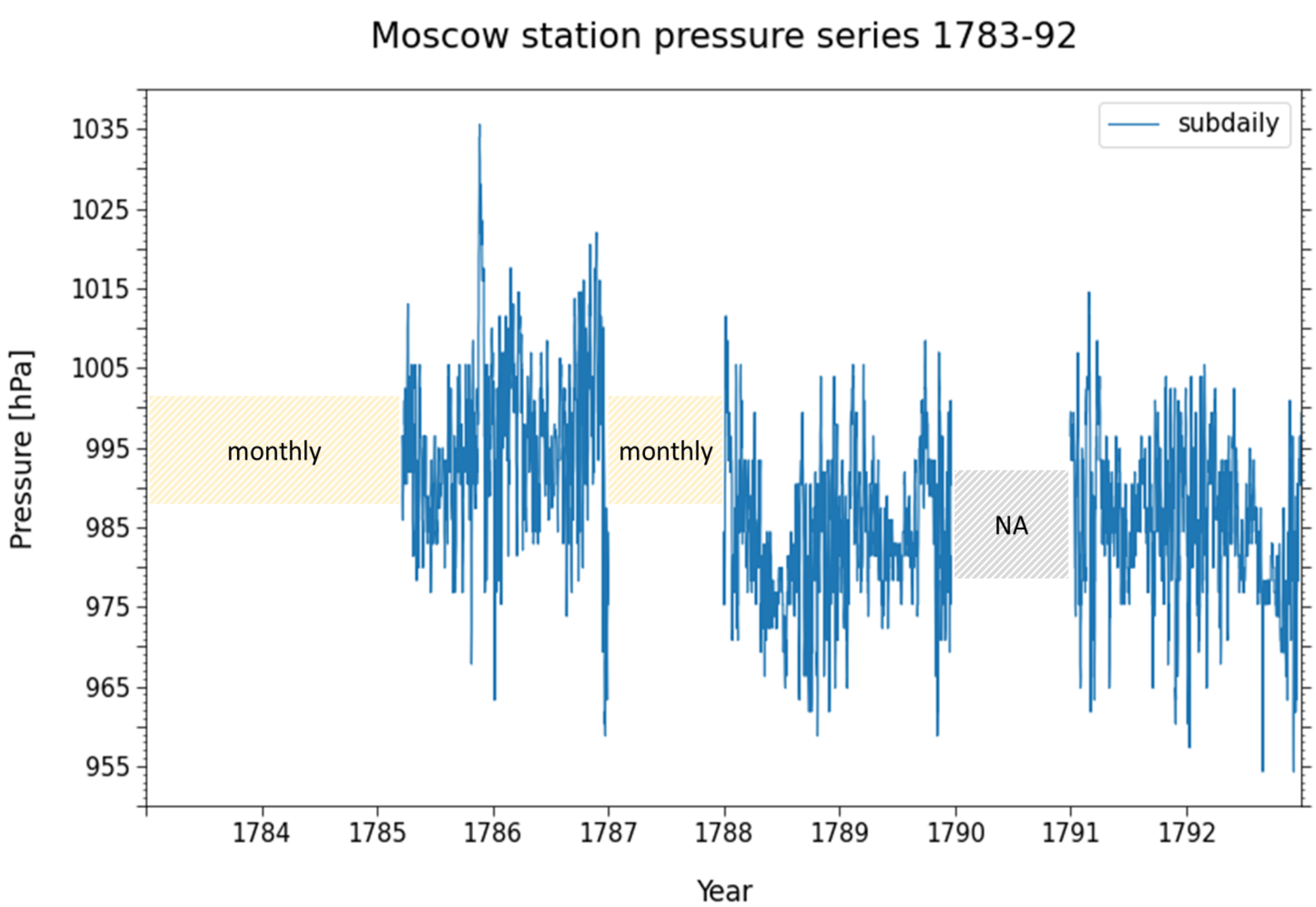

**Figure 4.** Station pressure series of Moscow in the period 1783-92. Shown are subdaily values in hPa (blue) and shadings that represent the availability of monthly data (yellow) or no available data (gray).

Keeping in mind the usual degree of uncertainty associated with early instrumental measurements, the temperature and pressure SMP data restored in this study are in usable condition. Beyond the problems discussed in this Section, they generally display very few errors or large issues that cannot be overcome.

### 2.6 Homogenisation

Homogenisation is a final step typically performed to identify and remove systematic biases – i.e. non-climatic influences – that may have arisen in the climate time series. This can be due to a change in the local environment, the instrument, the observer, or perhaps even a change in observational procedures. Temperature and pressure series of 7 years and longer were thus tested for homogeneity visually by applying the Craddock test (Craddock, 1979). This involves testing each series against other series based on their monthly cumulative differences according to the following equation:

$$c(t_i) = c(t_{i-1}) + \bar{x}_c - \bar{x}_r + x_r(t_i) - x_c(t_i) \tag{7}$$

with $c(t_0)$ equal to zero and where the candidate series to be tested is $x_c(t_1...t_n)$ and the reference series is $x_r(t_1...t_n)$. If either series is inhomogeneous – i.e. their mean has a sudden change – $c$ will exhibit a sharp change in slope. It is useful to plot $c$ several times against mutiple well-correlated reference series in order to better detect the inhomogeneity (or breakpoint) and to avoid making bad corrections (see Fessehaye et al., 2019). According to Venema et al. (2012), compared to other breakpoint detection tests the Craddock test allows for good accuracy, though the manual effort involved is higher, meaning it is better suited to smaller datasets such as the one in this study. Furthermore, there is some subjectivity in the selection of breakpoints: the choice to homogenise or not is linked to the specific approach of the researcher (Brunetti et al., 2006).

Overall, the correlations in the test showed good agreement for both variables between stations in Central Europe. The SMP temperature series did not manifest any significant breakpoints, though the Craddock test allowed for the detection of other systematic quality problems: series such as Rome and Regensburg showed evidence of an annual cycle in the test, indicating that the thermometers may have been measuring indoor or at least in conditions of poor exposure; additionally, the Andechs series seems to overestimate temperature after spring 1791. The criteria for adjusting the pressure series was for the mean difference before and after the breakpoint to be greater than 1 hPa. The following pressure series were corrected at the breakpoints by applying a constant correction: Andechs, Bologna, Brussels, Erfurt, Gotthard, Hohenpeissenberg, Prague and Rome (see Table 2). These corrections were computed by calculating the mean difference between the candidate series with the reference series before and after the breakpoint. For instance, if the difference of a given pressure series with respect to the reference is on average 3 hPa before the breakpoint and 7 hPa after, the correction is to be 4 hPa. The more reference series are available to calculate this mean difference, the more accurate the estimated correction will be. No action was taken on the Russian stations due to the lack of suitable series and their relative inferior quality should be kept in mind for any analysis.

**Table 2.** Station pressure series for which breakpoints were detected, showing the location in the series where the breakpoint is, the extent of the correction, and where in the series this was applied.

| Series | Breakpoint(s) | Correction [hPa] | Application |
|---|---|---|---|
| **Andechs** | Jun 1782 | - 1.2 | 1 Jan 1781 - 30 Jun 1782 |
| | Mar 1787 | + 2.4 | 1 Jul 1786 - 31 Mar 1787 |
| **Bologna** | Aug 1783 | + 8.6 | 14 Jan 1782 - 23 Aug 1783 |
| | Dec 1784 | + 3.9 | 16 Jan 1784 - 31 Dec 1784 |
| **Brussels** | Jul 1782 | + 3 | 1 Jan 1782 - 31 Jul 1782 |
| | Dec 1783 | - 2.5 | 1 Dec 1783 - 31 Dec 1785 |
| **Erfurt** | Dec 1782 | + 3.2 | 1 Jan 1781 - 31 Dec 1782 |
| | Jan 1788 | + 2.4 | 1 Jan 1788 - 31 Dec 1788 |
| **Gotthard** | Mar 1782 | + 5.9 | 1 Jun 1781 - 31 Mar 1782 |
| **Hohenpeissenberg** | Dec 1788 | + 2.7 | 1 Jan 1788 - 31 Dec 1788 |
| **Prague** | Jul 1784 | + 4.3 | 1 Aug 1781 - 31 Jul 1784 |
| **Rome** | Dec 1783 | + 6 | 1 Jan 1782 - 31 Dec 1783 |
| | Oct 1788 | + 3.8 | 1 Oct 1788 - 31 Dec 1792 |

### 2.7 Generation of daily and monthly series

For temperature, the calculation of daily means from subdaily measurements is weighted taking into account the known observation times to represent the true means with respect to the UTC day. The SMP daily means of 1787/8 were also adjusted using the known observation times from the closest year for which these are available. This adjustment was done using monthly mean diurnal cycles of 2-metre temperature at the nearest grid point in the ERA5-Land reanalysis (Muñoz-Sabater, 2019; Hersbach et al., 2020), calculated over the period 2001-2018. The choice of the type and length of the reference diurnal cycle can have different effects on the adjustment of daily means. Thus, for comparative purposes, a corrected version of the Żagań temperature series from Przybylak et al. (2014) is available in the data supplement, which used hourly temperature data from Wrocław for the period 1999-2003. The conversion from local time to UTC was done assuming mean solar local time (i.e., the UTC offset is linearly proportional to longitude). We discard the seasonal fluctuations related to the apparent local solar time (as measured by a sundial), as these have a negligible effect on the daily means (see Camuffo et al., 2021). For pressure, the average of the available observations for a given day was calculated, with no correction for the diurnal cycle. Values that were flagged with the QC software are ignored in the computation. From the daily series, monthly averages were calculated, excluding months with more than 5 missing days; this was primarily to carry out the homogenisation tests discussed above. The monthly 'means' in the *Ephemerides* are arithmetic means of all subdaily measurements until around 1784, after which mid-range values between maxima and minima were used. It would seem that editors in Mannheim could not keep up with the

vast quantity of observations. Thus, monthly observations after 1784 were not included in the final monthly series.

## 3 Inventory

### 3.1 Summary

The *Ephemerides* entailed a lot of patchwork by Hemmer's team in Mannheim, who were clearly interested in including as much information as was reasonably possible. They included a lot of information loosely related to meteorology but not limited to it, always serving the interest of economic and cultural betterment. Our inventory has filtered out a lot of this information, and focused mainly on early instrumental measurements. As mentioned in Sect. 2.2, the inventory was constructed to display the availability of observations for most parameters, mainly meteorological ones – excluding the description of the state of sky and form of clouds. But the inventory omits any astronomical, phenological and demographic observations. It thus contains basic information about the availability of measurements for the 37 stations that participated in the project.

For each station, and for each year of the 12-year period, the inventory shows which parameter was being measured; if data is available, then the box is marked with an 'X'. The last four columns contain information about the resolution of the data (subdaily, daily or monthly), the observer or correspondent who took/oversaw the observations, any comments, and the page range from the corresponding *Ephemerides* volume where the observations in the inventory can be found. In some cases a variable will be available at a subdaily resolution and as monthly summaries; here, the highest available resolution is reported.

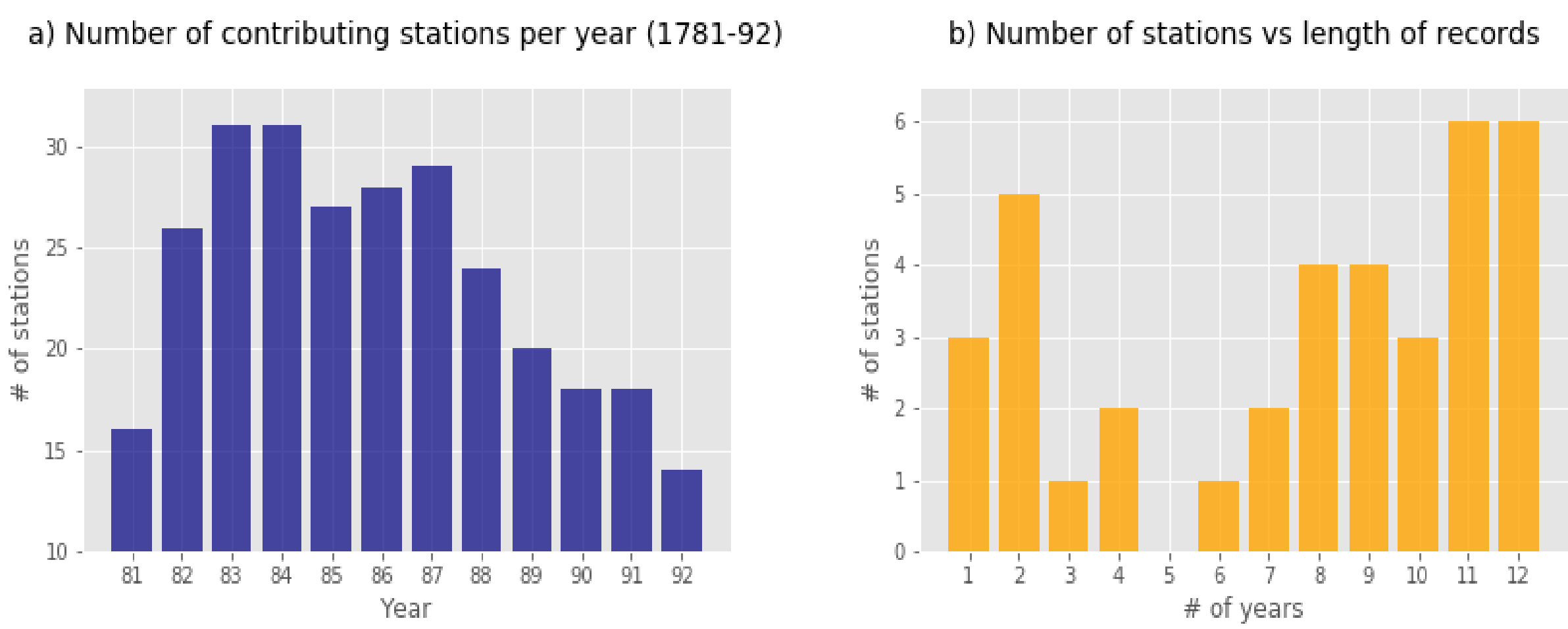

**Figure 5.** Bar charts showing (a) the number of stations for each year in the period 1781-92 and (b) the number of stations based on the length of available records.

At first glance, the inventory gives a rough idea of which stations were more prolific and which only participated for a few years; this information is summarised in Figure 5. The years with most participants are 1783 and 1784, which is fortunate for climate research as these years coincide with the Laki eruptions in Iceland, whereas the final years are those with the fewest participating stations. Indeed, 1788 marks the last year when a number of stations sent data to the Society, including dedicated stations that had provided observations for many years up until then, such as Berlin, Copenhagen, Erfurt, Middelburg, and Würzburg (see Table 1). A year later in 1789, Geneva and Tegernsee also stopped sending data. The last *Ephemerides* volume, for 1792, appeared in 1795, and it contained data recorded at 14 persevering stations.

Overall, more than half of the 37 stations in the network sent data to Mannheim for longer than 7 years; 12 of these, mostly stationed in Central Europe, had their data published in almost every *Ephemerides* issue. 11 stations provided observations for 4 years or less.

Cassidy's (1985) list of participating stations was designed to give a rough idea of which location participated in which year, but the list does not distinguish between the type or quantity of data. A station that in a specific year contributed only subdaily wind force observations, and in another year submitted subdaily measurements for every parameter, have a different value to a researcher. While the station did participate in both years, the weight of the contribution is different. Table 3 shows an extract from the inventory to illustrate this point.

**Table 3.** Extract from the inventory for the station in Budapest for the years 1786 to 1788, showing how distinguishing between which parameter observations are available is both useful and informative.

| **Budapest** | **1786** | **1787** | **1788** |
|---|---|---|---|
| Barometer | X | | X |
| Thermometer (indoor) | X | | |
| Thermometer (outdoor) | X | | X |
| Hygrometer | X | | X |
| Declinometer | X | | X |
| Wind direction & force | X | X | X |
| Rainfall | X | | X |
| Evaporation | | | |
| River height | X | | X |
| **Resolution** | subdaily | subdaily | daily (wind: subdaily; rainfall & river h.: monthly) |

Although there is a clear overarching framework that guided the collection of observations for the SMP, there is a number of lacunae and inconsistencies. Sometimes measurements were not taken as instructed by the SMP. For example, in the case of Żagań, measurements were not taken at regular intervals. In the case of St Petersburg, temperature and pressure were even observed at different times. Furthermore, not all observers used the instruments provided by the Society; for instance, the station in Cambridge, Massachusetts, used a Fahrenheit thermometer until 1784. In fact, the Russian stations seemed to follow a protocol of their own: Moscow and St Petersburg only provided pressure (likely already reduced to a standard temperature), outdoor temperature, and wind observations. Moreover, there is clear evidence that they switched instruments throughout the 12-year period. Moscow used a Delisle scale thermometer until 1787, then began reporting in Réaumur units; it also used a barometer in English inches, and eventually started reporting observations in Paris inches, though as mentioned in Sect. 2.5, it is unclear whether a SMP barometer was used. St Petersburg switched back and forth between Réaumur and Delisle thermometers. All these cases show why inconsistencies could arise.

As the inventory shows, it is also often the case that observations are only available for portions of the year, or that measurements of some parameters are interrupted or missing. In some cases, measurements are printed in the next edition of the *Ephemerides* as an *additamentum* – an addition – most likely due to the data not arriving in Mannheim in time to be organised, collated, and printed. For instance, Stockholm's subdaily tables from 1788 to 1791 are published together in the *additamentum* of the *Ephemerides* published in 1793 (covering 1791 data).

One particular pattern that emerges in the inventory is a 'thinning out' of measurements available for 1787 and 1788. The data received from each location for these two years was not printed *in extenso*, as they had been in the other *Ephemerides*. Wind direction and force observations are subdaily, but the rest of the data appear in the Appendices as daily or monthly means – one need only glance at Mannheim or Rome in the inventory to see this. Figure 6 illustrates this pattern very clearly. After these diluted editions in the late 1780s, the old extensive publication style resumes until the last volume, containing data for 1792, although by this time the size of the network is notably smaller.

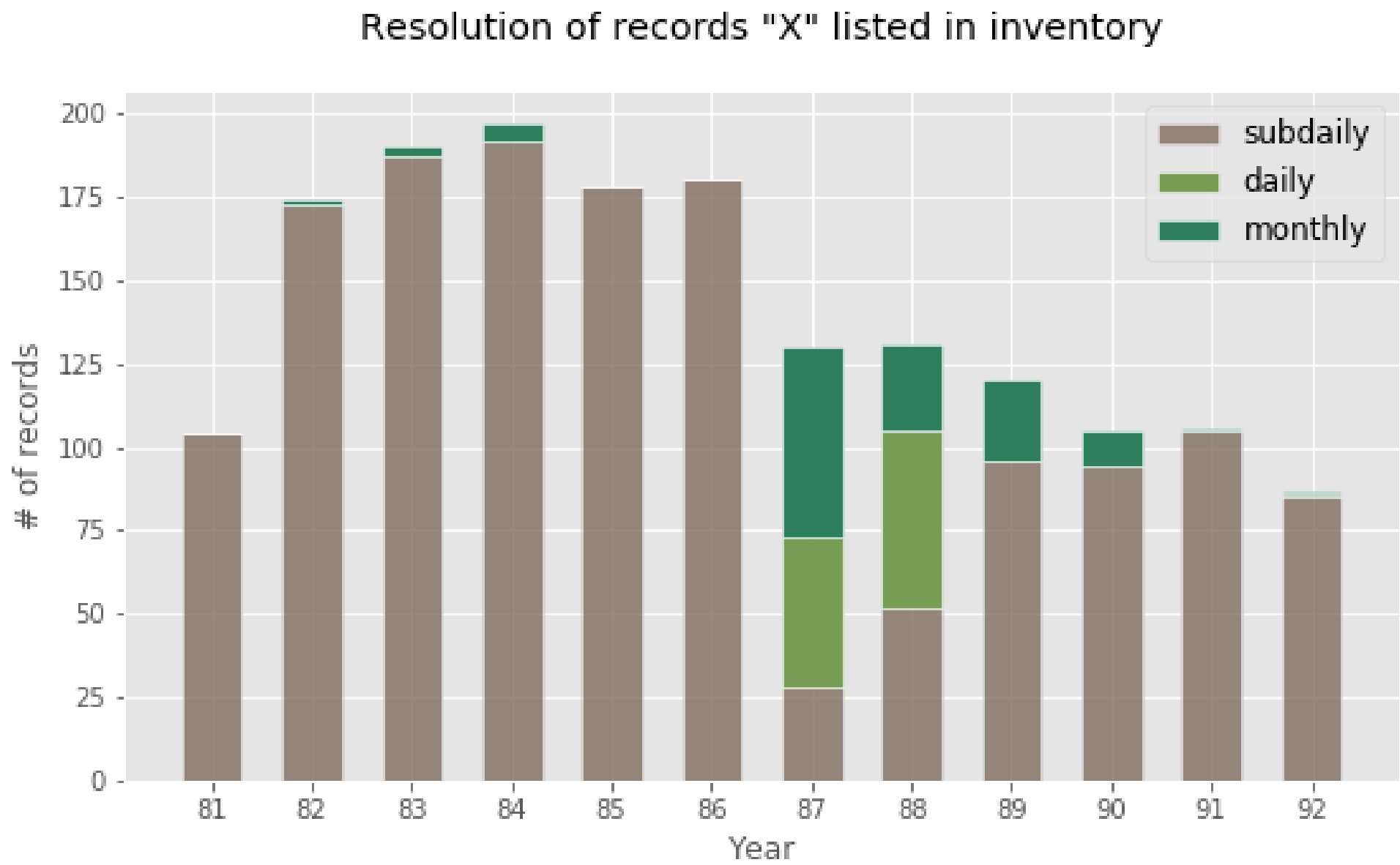

**Figure 6.** Bar chart showing the resolution for each data series marked by an 'X' in the inventory. The different formatting of the years 1787 and 1788 meant many records were printed as daily means and monthly summaries instead of the usual subdaily tables.

## 3.2 Discussion

There is an underlying structure and unifying theme to these 12 volumes, though the content of the *Ephemerides* is more heterogeneous than the literature claims. The protocol and instructions set out by the SMP were mostly followed, but not without the odd departure. The Russian stations and Cambridge (US) initially used their old instruments and did not receive their Réaumur thermometers until years later; in fact, delivering the instruments intact to these remote locations represented an enormous logistical hurdle. That these stations used different instruments with different scales may at first seem to go against the idea of having a robust network of comparable standardised instruments, but variation was not prohibited. Some instruments, such as the wind vane, hyetometer (rain), and atmidometer (evaporation), were actually described in the *Ephemerides* for local construction (1783, pp. 80-90). One downside to this, for instance, is that rainfall and evaporation were then often measured using local instruments and scales (sometimes unknown).

The perfection and improved construction of instruments was of the utmost importance to observers. Stationed in Marseille, Guillaume de Saint-Jacques de Silvabelle was keen to warn readers that the barometer initially sent to him by the SMP did not contain enough mercury and as a result 1782-3 pressure measurements were 3 lines too high; this issue was fixed in 1784 (*Ephemerides*, 1785, p. 498). So, as long as the observers' own instruments were in very good accord with those of the SMP, they could be used. This practice is known to have been the case for Dijon and St Petersburg, which also sent data measured with their own old devices (Euler, 1782; Maret, 1783).

The inventory makes clear that not every station belonging to the network made measurements for the whole period, nor did they always record the observations in a consistent manner at a subdaily resolution, at the specific hours instructed by the SMP. That said, the observers had been carefully selected by the SMP from academies, monasteries and universities across Europe to ensure that observations would be taken even after the death of initial observers (Cassidy, 1985). Indeed, it was Hemmer's decision to target organisations rather than individuals that contributed to the SMP's collaborative success. Despite the guaranteed continuity of the observations, the gathered measurements were still a product of the specific observer and therefore susceptible to heterogeneity.

The scientific dedication and participation of like-minded people was crucial to the project. The personalities and interests of SMP observers would have affected the quality of the series. Taking observations in Bologna, for instance, Abbot Petronio Matteucci was far more interested in astronomy than meteorology and was therefore not too diligent in organising the meteorological observations for the SMP (Camuffo et al., 2017). He prepared subdaily observations for 1782-4, 1791-2, daily means for 1788, and substituted subdaily readings for monthly averages in 1789 and 1790; 1785 and 1786 were skipped alltogether. The observations include at times significant gaps, also the pressure series needed homogenising. Almost in stark contrast to this example, reporting from Padua, were Giuseppe Toaldo and his nephew Vincenzo Chiminello. Both were good and accurate scientists – especially Chiminello, whose handwriting was neat and ordered (Camuffo et al., 2020). This commitment is reflected to some extent in the inventory: Padua followed (with few departures) the protocol set out by Hemmer and became one of the few stations that reported measurements for every year of the 12-year period. Indeed, unlike for Bologna, the recovered temperature and pressure series for Padua are in very good condition and required no adjustments.

Decisions by the SMP itself in Mannheim were also a factor affecting the year-to-year changes in the *Ephemerides*: the edition published in 1790 – containing the observations for 1788 – marks not only the absence of a number of stations that had hitherto been reporting measurements, but was also the year with the highest proportion of lower resolution observations. There is not much information in the *Ephemerides* as to why the volumes containing data for 1787 and 1788 were not published in extenso. The preface to the 1787 *Ephemerides*, published in 1789, mentions that the SMP is glad to finally be able to publish the volume after having overcome various "obstacles" posed by editing (*Ephemerides*, 1789, p. v). It would seem that there were delays due to observations from various stations arriving late and at different quality/resolution, which may have caused the decision to adopt this abridged format. The preface for the 1788 volume, published in 1790, complains about the strains of editing and compiling getting "higher every year" (*Ephemerides*, 1790, p. v-vi); this also may explain the choice to simplify the format then. Nevertheless, it was decided to revert to the old unabridged format after "many" expressed their "dissatisfaction" (*Ephemerides*, 1790, p. v). To today's climate scientist wishing to use these data for research, these diluted editions should not become an issue, as pressure, temperature, moisture and other data are usually still available as tables of daily means, or monthly maxima and minima. That said, monthly data that has been calculated from quality-controlled subdaily observations would lead to a higher quality monthly product than that obtained from the mid-range values of the monthly tables in the *Ephemerides*.

There is another issue for researchers who wish to use this data to reconstruct meaningful synoptic weather maps. That bigger issue is the gradual loss of synoptic coverage over Europe after 1787-9, as stations stopped sending observations to

the SMP. This is made more problematic by the fact that most of the station observations in 1787 are in the form of monthly maxima, minima, and mid-range values. However, even this shortcoming can be overcome by supplementing the analysis with data from independent neighbouring stations that were recording observations at the time. That thus ensures better geographic distribution – though this may be harder to achieve in Eastern Europe.

Retrospectively, the declining participation among stations is no surprise. Jakob Hemmer died unexpectedly in 1790 without designating a successor (Cassidy, 1985, p. 19): the impetus of the SMP's project had thus died with its leading organiser. Furthermore, what had been difficult obstacles to overcome up until then, such as the shipment of fragile instruments, personal rivalries, and poor communication, became compounded by the rapidly changing political situation in Europe. Hemmer himself had complained about the extreme difficulty in delivering instruments to the stations, which sometimes arrived vitiated and in pieces due in part to the negligence of custom officials, especially in France (*Ephemerides*, 1983, p. 45). It is no wonder that most of the stations that delivered data for the entire twelve years were in Palatinate-Bavaria. In 1792, France declared war against Austria and started its eastward expansion, occupying Mannheim on its way and putting an abrupt end to the Society's activities. One publication even mentions that on 8 November 1792 the observer for the Brussels station, the Abbot Théodore Augustin Mann, had been forced to escape Belgium because of the French invasion (*Ephemerides*, 1795, p. 112). The French Revolution undoubtedly weakened the SMP's otherwise efficient integration of the network.

Despite the few inhomogeneities highlighted in the inventory, it remains a fact that the SMP managed to integrate and coordinate the collection of meteorological data. Amazingly, they did it rather efficiently at a time when this sort of scientific communication was very difficult, compounded also by personal and political differences. It is only due to the relatively tight control of its members, long-term central coordination, and perseverance of the SMP's founder, Hemmer, that this project lasted as long as it did. Thanks to the efforts of the Society's member stations, today we have access to unique climate data for the late 18th century.

## 4 Examples

### 4.1 Storms during the winter 1783/84

In this example we look at some of the recovered SMP pressure data to explore some of the severe storms from the Northwest that hit Europe from late December 1783 to January 1784 (see Lamb, 2005, pp. 87-8). While the greatest impacts from the storms – in terms of destructive winds – occurred in the British Isles, the severe flooding that ensued in parts of mainland Europe during this season have been well-documented (Demarée, 2006; Brázdil et al., 2010). There was considerable material damage in wide regions north of the Alps that exacerbated the already harsh socio-economic conditions caused by the severe winter.

These storms occurred following a long spell of extreme weather. Europe had been experiencing bitter cold temperatures, the result of persistent high pressure systems that had lingered over Central Europe from 26 November to 16 December, which were then followed by successive northerly and easterly winds – as evidenced from the CAP7 weather type reconstructions

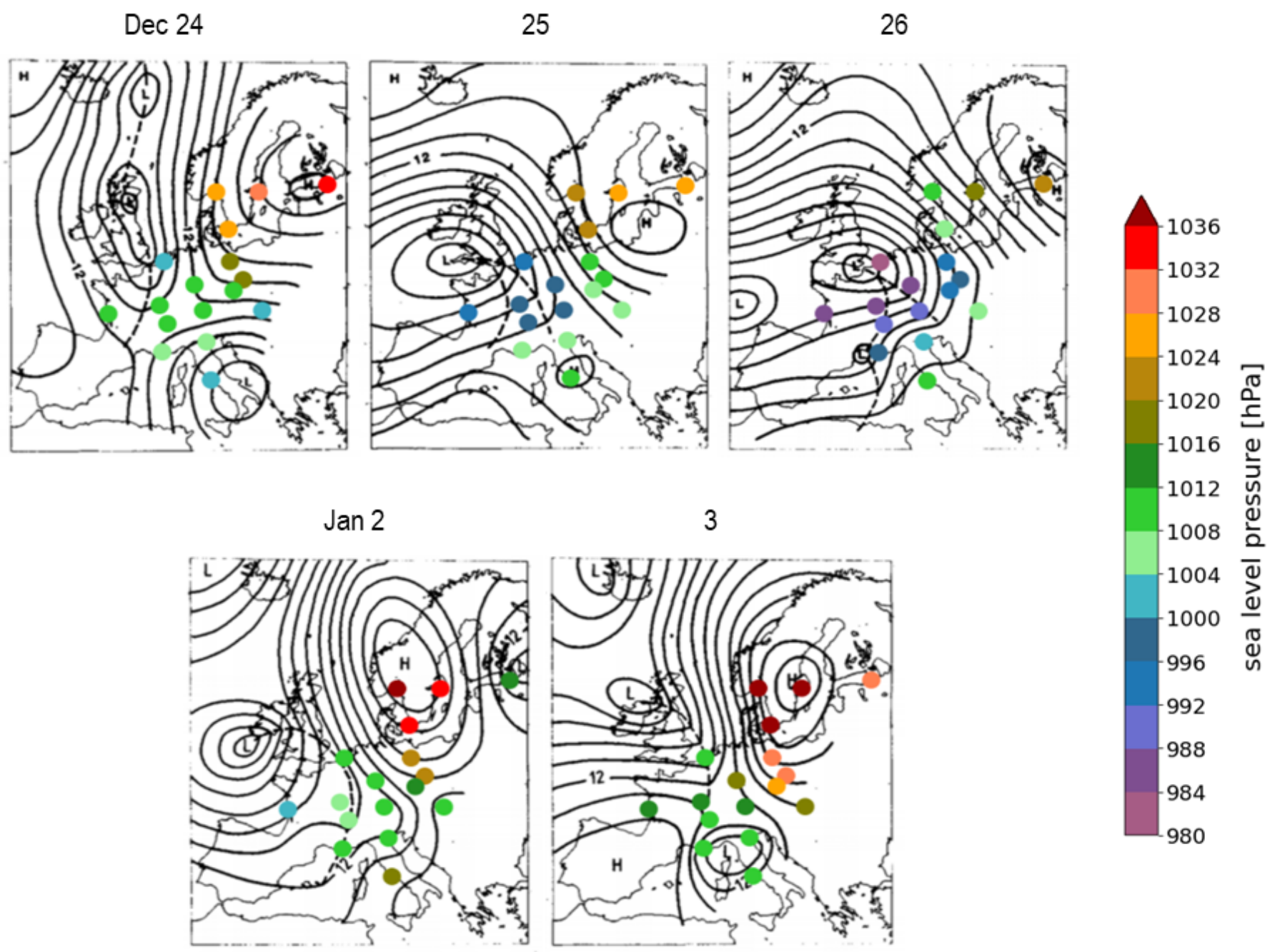

**Figure 7.** Kington's hand-drawn synoptic maps showing the storms as they moved over Europe covering the days 23-26 December 1783 and 2-3 January 1784. Isobars are drawn at 4 hPa intervals, and '12' denotes the 1012 hPa isobar; dashed lines indicate fronts. Coloured points represent SMP pressure observations in hPa reduced to sea level from 17 selected stations. Adapted from Kington (1988, pp. 99-100), reproduced with permission of Cambridge University Press through PLSclear.

by Schwander et al. (2017), which extend back to the year 1763 MeteoSwiss' CAP9 weather type classification (Weusthoff, 2011). All this was acting to prevent milder temperatures from the Atlantic Ocean to move eastwards over the continent. With the storm around Christmas came a slight upswing in temperatures and with it heavy rainfall and snowmelt that caused the first floods; this was the case for the Dijle and Demer catchments of the Scheldt basin in Belgium. The Gete River also flooded at Orp-le-Grand on 2 January and at Tienen on 4 January 1784 (Demarée, 2006). In Mannheim, Hemmer noted how the rivers "swelled greatly"; the thick ice that had constrained the Neckar tributary of the Rhine rose and broke; around 6 in the evening the river flooded at Feudenheim, inondating the bridge and breaking parts of it off, which flowed all the way downtown towards the station (*Ephemerides*, 1786, p. 53).

John Kington's hand-drawn synoptic maps (Kington, 1988) are a helpful source that allow a unique insight into the daily progression of weather over Europe during this event. Not surprisingly, the SMP data fit very well with Kington's daily SLP fields. Figure 7 shows SMP pressure data reduced to sea level (using Eq. 6) plotted onto Kington's maps. By 24 December a depression had arrived from the Northwest and soon developed an intense centre that within days had moved over the English Channel. As the low expanded all sites recorded a drop in pressure with Middelburg and Mannheim respectively at 984 and

986.2 hPa. On 2-3 January another frontal system arrived in from the Atlantic, pushing against a large high pressure system centred over Scandinavia, creating a strong pressure gradient over the North Sea that blew gusts of Arctic air over Denmark, Germany and the Low Countries. According to SMP measurements, the tight gradient ranged on 2 January from 1000.2 hPa in La Rochelle to 1037.2 hPa in Spydeberg.

This brief example demonstrates the reliability of the processed pressure data as well as its usefulness when paired with documentary sources. These measurements could thus be used to compute statistical reconstructions that would allow us to better understand such extreme events.

### 4.2 1785: the coldest March

In Central Europe, March 1785 may well have been the coldest March in over 300 years. According to the EKF400v2 reanalysis (Franke et al., 2017; Valler et al., 2021), this was the coldest March since 1701 for the regions surrounding Mannheim and Padua, the second coldest for the region in and around Żagań, and for Budapest it remains in the top ten coldest. Table 4 shows the extent of these anomalies: for Żagań, March 1785 was over 6°C colder than the average during the period of the SMP's activity, and around 5°C colder for both Mannheim and Padua. There is reasonable agreement between the SMP monthly temperature anomalies and those of the EKF400v2, with a difference <0.7°C for all stations, except for Budapest (with a difference of 2°C), meaning that average March temperatures 1781-92 for this location are higher for the SMP observations than in the reanalysis. These differences are partly due to the coarseness of the reanalysis resolution compared to the station data.

In Switzerland, the Vierwaldstätter Lake had frozen over in Lucerne's bay and similarly the Geneva Lake at the Geneva-end

**Table 4.** Monthly anomalies of March 1785 at four stations in Central Europe: Mannheim, Padua, Budapest, and Żagań. The first column is calculated with respect to monthly climatologies from the SMP instrumental measurements, whereas the other three columns are calculated with respect to the climatologies of the EKF400v2 reanalysis gridboxes closest to the location of the four stations. (*) anomalies computed excluding 1781 and 1787.

| | **anomalies [°C]** | | | |
|---|---|---|---|---|
| **Stations** | **1781-92 Palatina** | **1781-92 EKF400v2** | **1751-1800 EKF400v2** | **1951-2000 EKF400v2** |
| Mannheim | -4.8 | -5.3 | -6 | -6.5 |
| Padua | -5.1 | -4.4 | -5.2 | -4.7 |
| Budapest | -5.6* | -3.6* | -3.6 | -4.1 |
| Żagań | -6.7 | -6 | -6.5 | -6.8 |

(Pfister, 1999, p. 121). The SMP observer stationed in Geneva, Jean Senebier, had been observing subzero temperatures for days and described how by 2 March the lake had frozen over by the mouth of the Rhône (*Ephemerides*, 1787, p. 215). The future 2nd President of the USA, John Adams, working as 1st Minister to the Netherlands and living near Paris at the time, wrote about his conversation with a local abbot, recording "that neither he nor his brother, (and they are both turned of seventy) remember ever to have experienced so cold weather in the beginning of March" (Adams, 1785/1981).

SMP data allow us to track the daily development of temperature throughout these weeks. Shown in Figure 8 are mean daily temperature series for selected locations, along with the highest probability daily weather types; the series have very good agreement, displaying mutual correlations between 0.78 and 0.91. Two features are immediately apparent. The first is the severe cold snap from 27 February to 2 March, with Prague and Żagań reaching temperatures below -20°C. This was likely due to the strong Northeast winds – type 1 weather type according to Schwander et al. (2017) – bringing freezing continental air into Central Europe and was soon reversed by milder West-Southwest winds (type 2). The second feature is the cold March itself, manifesting for many locations as persistent below zero conditions for almost the entire month. Looking at the weather types, the main cause seems to have been successive days of type 4 (East, indifferent), type 1 (Northeast, indifferent), type 5 (High pressure) and type 6 (North), hinting to the constant flow of cold air into the region.

Both the Palatina series and the weather types are consistent with Kington's daily synoptic maps. The maps show blocking

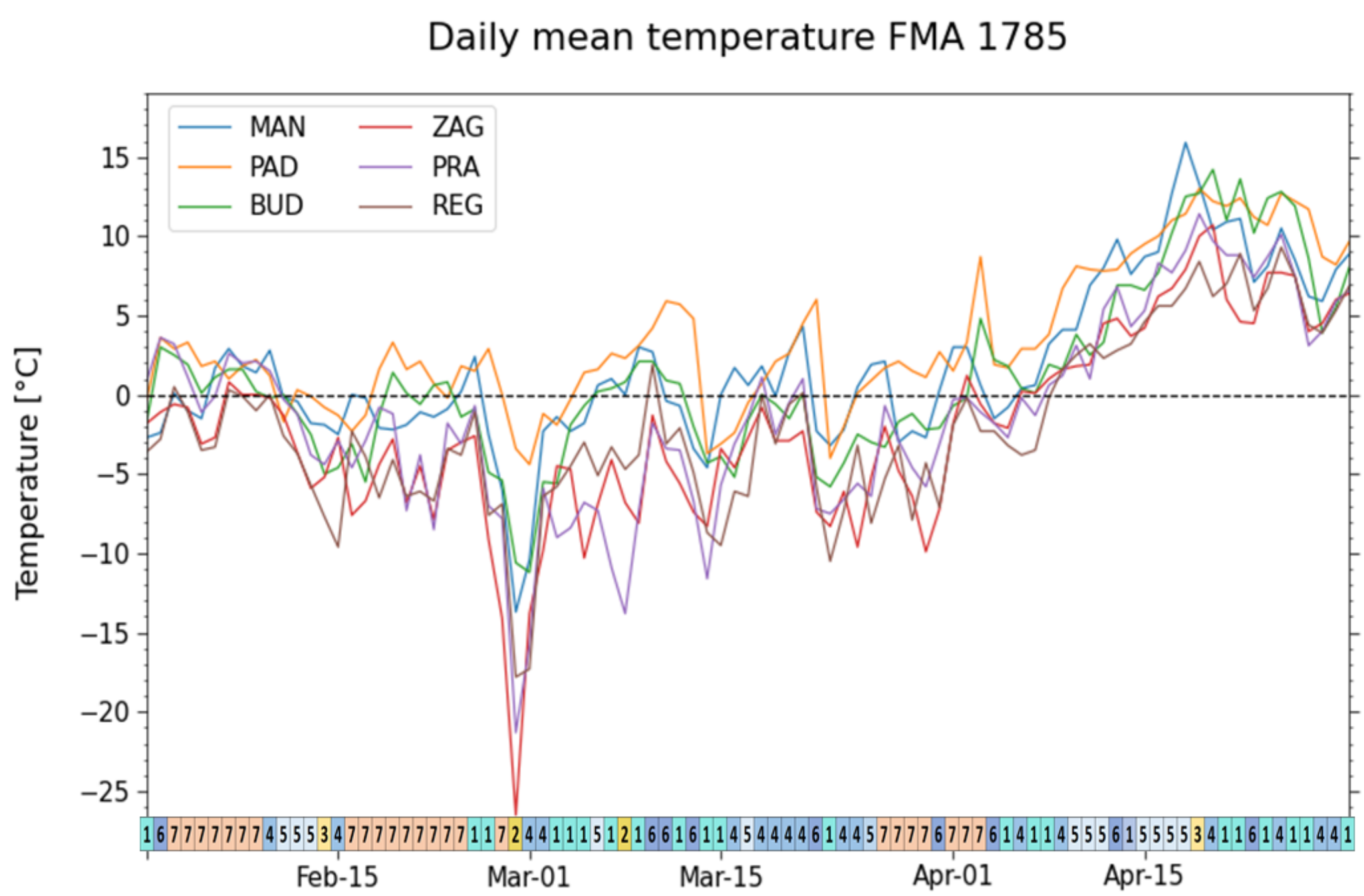

**Figure 8.** Daily mean temperature from February to April 1785 at six locations in Central Europe: Mannheim, Padua, Budapest, Żagań, Prague, and Regensburg. The bottom bar indicates the CAP7 weather type classifications by Schwander et al. (2017).

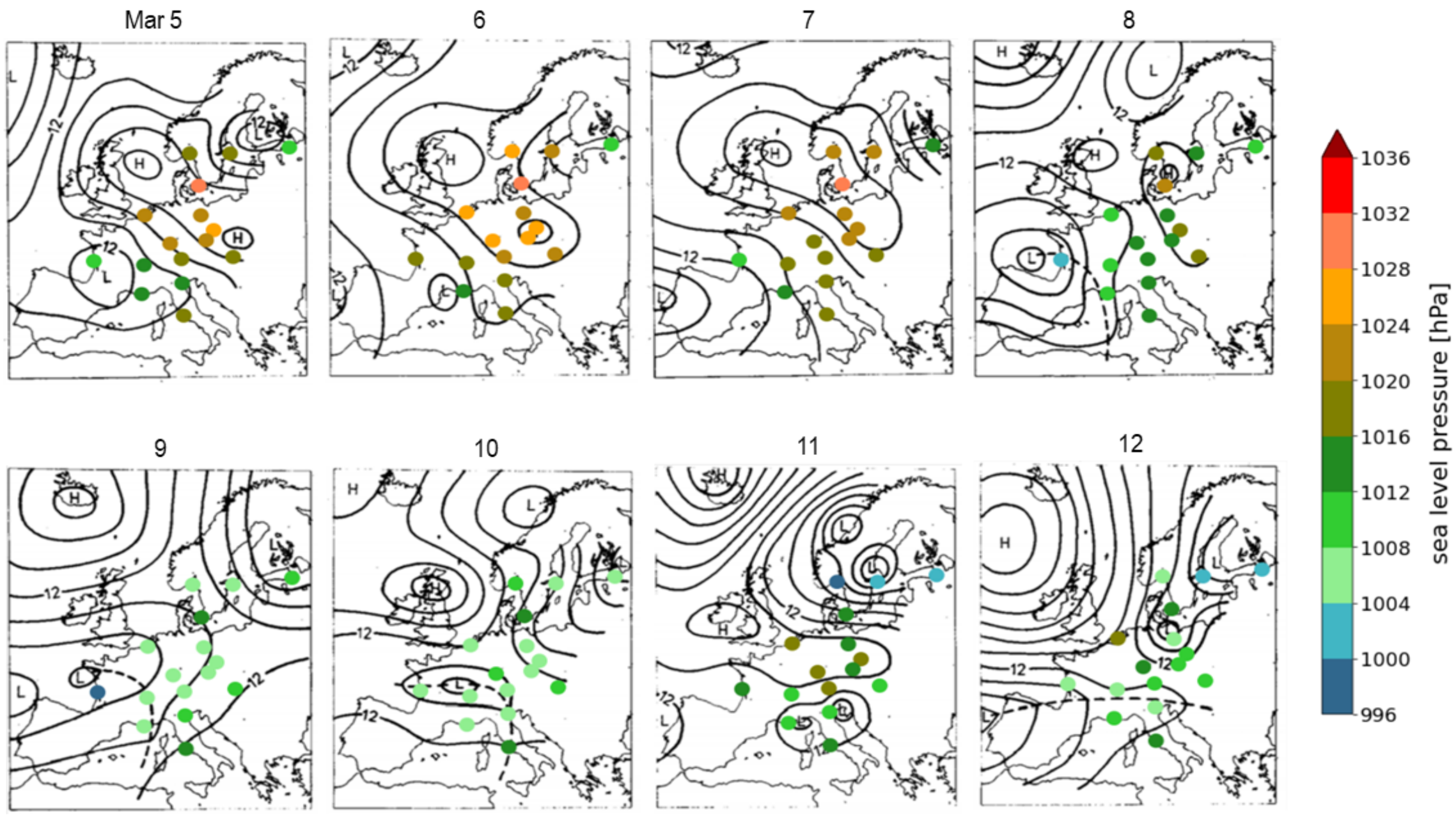

**Figure 9.** Kington's hand-drawn synoptic maps showing main SLP patterns over Europe covering the days 8-12 March 1785. H and L denote respectively centres of high and low pressure. Isobars are drawn at 4 hPa intervals, and '12' denotes the 1012 hPa isobar; dashed lines indicate fronts. Adapted from Kington (1988, p. 128), reproduced with permission of Cambridge University Press through PLSclear.

over Northwest Europe until 7 March with a centre over the North Sea and Scotland (see Fig. 9); this is interrupted by a large cold front moving into France on 8 March, which coincides with a slight warming in the SMP series and a type 2 weather type; over the next weeks blocking resumes, this time mainly between Great Britain and Iceland (occasionally over Ireland), providing a constant flow of northerly Arctic air into Central Europe.

This example demonstrates once more the value of the SMP daily series recovered in this study. Using the observations in the *Ephemerides*, combined with the CAP7 weather types and Kington's invaluable maps, it is possible to get a detailed view of this extreme month. These data have great potential to explore other temperature extremes in late 18th century Central Europe.

## 5 Conclusions

In our study we examined some of the historical weather data from the *Societas Meteorologica Palatina* to fill a perceived gap in recent climate research. In a first part the relevant metadata and availability of instrumental records in the *Ephemerides* were compiled into an inventory. 37 temperature and pressure records were digitised, converted, corrected where needed, and formatted into an accessible and standard structure (SEF). Two cases of extreme weather were explored to highlight the potential

of SMP instrumental data in reconstructing 18th century weather.

The inventory reveals that the availability of records is heterogeneous and depends on the station and variable in question. The majority of the available records is at subdaily resolution, though the publications containing data for 1788-9 are predominantly daily and monthly averages. Most stations took measurements for more than 7 years, yet some others contributed only for 1-2 years. The inventory can be expanded and improved upon by focusing on specific stations or other variables, such as air moisture and rainfall.

The overall quality of the recovered temperature and pressure series is good, likely due to the SMP's insistence on the use and free dispatch of standardised, comparable instruments. Nevertheless, the series are not free of errors: a number of pressure series needed to be homogenised, and this could not be done for the Russian data; a number of temperature series occasionally showed evidence the instrument being badly exposed, either indoors or in the sunlight. Suspicious values were flagged with the C3S QC software, thereby providing an indication of the kind of problem at hand. The uncertainties associated with these measurements generate a need for locating and exploring further details regarding the individual stations.

The two examples demonstrate the value of the instrumental data, especially when paired with documentary sources and weather type reconstructions. These records have a number of different possible applications for climate research, including their assimilation in reanalysis datasets for use in regional dynamical downscaling studies and their use in the statistical reconstruction of daily fields over Europe with the analog method (see Pfister et al., 2020). Their usefulness may depend on the specific goals of the study being undertaken, including which focus area and variables are of interest. Additionally, the specific uncertainties of a series should be taken into consideration when evaluating any results.

The potential here is large not only for climate science but for historians as well. The generated time series are of sufficient quality to allow historians to inspect the impact of specific periods of extreme weather on historical events. The weather information uncovered here could be used to study social developments that might have depended on day-to-day weather, or indeed on health and agriculture. More specifically, one could trace the daily impact of weather during this period on political and economic affairs, military operations, or any travels and expeditions. Databases of the type presented in our study will certainly benefit from the deep collaboration between historians and scientists.

*Code and data availability.* The inventory, data and relevant code related to this article are available as supplement.

*Author contributions.* DP created the inventory, processed the data, performed the analysis, created the figures, and drafted the manuscript. YB oversaw the digitisation and collection of measurements, as well as verified the data rescue process. SJ, AP, and RP contributed a number of digitised series and, together with CR, they provided critical feedback to improve the final version of the manuscript. SB helped shape the research and supervised the project.

*Competing interests.* The authors declare that they have no conflict of interest.

*Acknowledgements.* This work was supported by the Swiss National Science Foundation (project WeaR 188701), by the European Research Council (ERC) und the European Union's Horizon 2020 research and innovation programme grant agreement No 787574 (PALAEO-RA). SJ's work was carried out within the ANR CHEdaR Project (ANR-09-CEP-002). The work of AP and RP was conducted within the NCN project Nr DEC-2020/37/B/ST10/00710.

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
