# Peer review of "Unlocking weather observations from the *Societas Meteorologica Palatina* (1781-1792)"

_Climate of the Past, 2021_

## Author Comment (AC1)

**Response to the review by Anonymous Referee #1 on the manuscript cp-2020-57 "Unlocking weather observations from the Societas Meteorologica Palatina (1781-1792)" by Duncan Pappert et al.**

We thank the reviewer for their effort in carefully reading and commenting on our manuscript. In the following, we reply to their comments point by point.

*Review: The database will undoubtedly be of importance to historical climatologists, and this manuscript could be useful to users of the database and to historical climatologists in general. However, as explained below, the article would be much more useful and impactful if the structure were revised for clarity. In particular, the authors should distinguish discussions and analysis of the original SMP observations from discussions and applications of the new database. Finally, the manuscript should address specific questions and concerns at the end of this review.*

*First, this structure can be confusing for the reader and makes it difficult to find and remember important content. Some topics are divided across multiple sections, while in other cases information has to be repeated. For example, sections 2.1 and 3.1 substantially overlap, and it's not clear why some information went in one section instead of the other. The fact that the original society reports contained much non-meteorological information and that this information was left out of the database seem like items that should have been discussed in "methods and data". On the other hand, much of the "methods and data" section consists of analysis or induction that I would normally expect in a "results" section (which is altogether missing from the article). These issues of clarity are problematic in an article such as this one, which is meant to serve as a guide to users.*

*Second, the structure does not allow the author to systematically address the project's context within, and contributions to, each field of research relevant to this study. These fields include:*

- *The history of meteorology and climatology;*
- *The study of early weather instruments and their measurements;*
- *Data rescue of early instrumental measurements;*
- *The creation of useful historical climatology databases; and*
- *High-resolution historical climate reconstruction.*

*I wouldn't expect detailed discussion of each of these. However, simply keeping them in mind and addressing each specifically in the introduction and conclusion would help frame the article and ensure it does not neglect important context or applications.*

*Third, the current structure blurs discussion and analysis of the methods and publications of the original SMP observation network and the usefulness of its measurements, on the one hand, with discussion and analysis of the methods and organization of the new database and its usefulness for climate reconstruction, on the other. Each of these is an interesting subject in its own right. As a researcher in historical climatology and potential user of the database, I would like to know (1) about the SMP, its background, and its activities; (2) about the new database and how it works; (3) what the database can tell us about the SMP, its coverage, and the quality of its observations; and (4) whether and how the database can help improve climate reconstructions. While that information is present in various places in the manuscript, it is currently all mixed together in ways that make it very hard to get at a good distinct explanation of each of these topics.*

*As referee, I would not want to dictate the structure of the article. There are various ways the authors could revise or amend the current structure to address the three major issues outlined above. However, I suggest the following possible changes:*

- *The introduction could address the project's context within, and contributions to, the relevant research fields, as described above. The background and history of the SMP would fit better in the section on sources and methods.*

- *The section on sources and methods could separate (1) the structure, methods, and publications of the SMP from (2) the structure of the database, its methods of compiling SMP data, the methods used to analyze the SMP network with the database, methods of correcting and homogenizing SMP measurements as they are put into the database, and how the database could be compared to other data or studies to examine its value for reconstruction.*

- *There could be a "results" section presenting (1) the results of the analyses of SMP data for coverage, resolution, and consistency (currently in section 3.1); (2) results of analyses of tests for homogeneity and breakpoints (currently in section 2); (3) results of comparisons with other data or reconstructions (currently in section 4).*

- *A "discussion" section could include much of the discussion in section 3.2 and the conclusion.*

- *The conclusion could recap the article and address the project's contribution to the relevant fields of study presented in the introduction.*

**Reply:** The current structure of the paper was meant to reflect the dual goal of the study: 1) to create an inventory of meteorological observations carried out by the Palatina network ; 2) to digitize and correct the temperature and pressure observations, as well as demonstrate their potential usefulness. Perhaps the attempt to separate these two 'projects' has come at the expense of the more traditional 'methods-results-discussion' structure and resulted in a somewhat scattered and confused structure. The data rescue/processing part in the methods section was explained in detail and so were some of the identified issues which were addressed there and then to create cohesion and avoid mentioning these in a later part (e.g. in a 'results' section as suggested by the referee): this way, for instance, the 'why' 'what' and 'how' of homogenization is covered in one subchapter.

The 'Inventory' chapter is meant to be a results section dedicated to the first of the two study goals. The first part 'Summary' summarizes the main features and findings of this cataloguing exercise, and the second part 'Discussion' (perhaps a misleading name) discusses more or less point by point the features presented in the summary. This 'discussion' refers solely to a discussion of the Inventory, which necessitates more targeted explanations about the background of the Palatine Society; it is not meant as a discussion for the whole paper.

The 'Examples' chapter stands as a results for the second goal and is meant to showcase the measurement series themselves. Understandably, this is confusing, as some of the 'results' of the measurements were addressed in the methods section.

As the structure is a concern raised by the second Referee as well, the revised draft of the manuscript will clearly need some rewriting as well as a reshuffling of the structure. This will be done following some of the advised suggestion listed above. The context of the paper and its contribution to different fields of research will be addressed better in both the introduction and conclusion.

Content pertaining to either Methods, Results and Discussion will be more clearly divided into these groups.

*Review: Is the database compatible with those of other early instrumental data rescue projects such as ACRE? Are the results being incorporated into ACRE or another such project?*

**Reply:** Two of the co-authors were directly involved in the establishment of the Copernicus Data Rescue Service, which is tightly related with ACRE, and collaborate regularly with ACRE members. The data format complies with the best practices recommended by Copernicus (https://climate.copernicus.eu/sites/default/files/2021-05/C3S_DC3S311a_Lot1.3.4.2_2020_BestPracticeGuidelines_Part2.pdf). The data will be eventually submitted to the Copernicus database (GLAMOD).

*Review: Have any of the original instruments from the SMP (or other examples of the same manufacture) survived? If so, have they been analyzed for any particular errors or biases, particularly those that would require non-linear corrections at high or low temperatures (in addition to the data processing already discussed in section 2.3)?*

**Reply:** We do not know of any instrument that survived, although certainly some must have. We visited the astronomical observatories in Geneva and Bologna (i.e., the two Palatina stations closest to Bern) but did not find any instrument from the Palatina in their collections. In any case, the instruments reflected the standard for the time and were already rather accurate (see, e.g., the literature from the IMPROVE project – Camuffo and Jones, 2002). In fact, reliable international calibration standards were already in place (Cavendish, 1777). The main instrumental error source that we are aware of is the contraction of the glass, and that is accounted for in our corrections. Another known problem of the Palatina thermometer is that it was only graduated until -16°Ré, which makes lower temperatures somewhat less accurate (we will add this information to the manuscript).
Larger, non-linear errors are probably related to the exposure of the thermometer, but they are hard to estimate and out of the scope of this paper, as they affect pretty much any temperature measurement up until at least the early 20[th] century.

*Review: The case study in section 4.1 compares information in the new database with the reconstructions of Kington 1988. However, as described in the introduction (line 55-56), Kington 1988's reconstruction is also based on SMP measurements. Therefore, the comparison does not justify the conclusion that "This brief example demonstrates the reliability of the processed pressure data as well as its usefulness when paired with documentary sources" (lines 426-427). For tests of reliability, the processed SMP data should be compared with other early instrumental series or high-resolution reconstructions based on reliable phenological observations. In fact, if there were differences between Kington 1988 and the new database, then that would actually underline the importance of the correction and homogenization applied to the SMP data. Instead, this example seems to illustrate relative ease of producing and visualizing data with the new database (as opposed to the laborious hand-drawn maps of Kington).*
*Similarly, in section 4.2, can the authors confirm that the EKF400v2 reanalysis (line 431) did not itself use SMP measurements?*

**Reply:** Yes, 'reliability' may be the wrong choice of words as it implies a validation against high-quality independent observations. This is not strictly the case here as both Kington's hand-drawn maps and EKF400 used SMP data (although in the latter case, only a handful of series were used). The examples in the study are meant to showcase the usefulness of the recovered series, their agreement with each other, and how they fit with and complement existing work done on this

period. Combined with information from historical databases, the potential of these measurements could be further uncovered.

As the main added value of this database in comparison with existing products is its daily resolution, there are no independent, high-quality measurements with that resolution that we can validate against.

*Review: The manuscript is generally well written but contains occasional minor issues of English usage (for example, in line 30 "spurn" should be "spur").*

**Reply:** Will be corrected and the manuscript will be scanned for more such mistakes.

Camuffo D., Jones P. (eds), 2002, Improved Understanding of Past Climatic Variability from Early Daily European Instrumental Sources. Springer, Dordrecht. https://doi.org/10.1007/978-94-010-0371-1_1.

Cavendish, H., Heberden, Alex-Aubert, Deluc, Maskelyne, Horsley, Planta, 1777, The report of the Committee of the Royal Society to consider of the Best Method of Adjusting the Fixed Points of Thermometers; and the Precautions Necessary to be Used in Making Experiments with these Instruments, Philosophical Transactions 67, 816–857 .

---

## Author Comment (AC2)

**Response to the review by Anonymous Referee #2 on the manuscript cp-2020-57 "Unlocking weather observations from the Societas Meteorologica Palatina (1781-1792)" by Duncan Pappert et al.**

We thank the reviewer for their effort in carefully reading and commenting on our manuscript. In the following, we reply to their comments point by point.

*Review: This study reported the digitizing and compilation of the Societas Meteorologica Palatina (SMP) weather observation, a network of 37 stations across Europe plus a couple in North America and Greenland coving the decade of 1781 and 1792. The quality of the SMP temperature and pressure observation data is then evaluated by using the C3S Quality Control software to identify outliers and visual inspection. The potential of the SMP reconstruction for climate research is demonstrated by two extreme weather events of the reconstructed period. The reconstruction is rigorously done and described in great detail. However, the manuscript, in its current layout, requires major revision before it could be considered for publication in this journal.*

*Specific comments:*

1. ***The structure of the manuscript is not logically sound.*** *For example, section 2.1 on "Source material description" and section 3 on "Inventory" could be combined and shortened by removing the repeating or loosely relevant information. Section 2.6 on "Homogenisation" and section 2.7 on "Generation of daily and monthly series" should be parts of the "Data processing (section 2.3)" work. Section 2.7 on "Generation of daily and monthly series" should also be placed before section 2.5 on "Quality control" and section 2.6, since a number of discussions in quality control and homogenization refer to the monthly data.*

**Reply:** As the structure is a concern raised by the first anonymous Referee as well, the revised draft of the manuscript will clearly need some rewriting as well as a reshuffling of the structure. To provide an explanation, the data and methods chapter was written trying to preserve the order in which the steps were performed. For instance, "Quality Control" was performed several times over at different stages of the data processing (e.g. before and after homogenization) but mostly at the subdaily resolution - that is, before calculating daily and monthly means; but in order to perform homogenization tests, monthly means need to be calculated from the quality controlled subdaily data. Your point that a number of discussions in quality control and homogenization refer to the monthly data calls attention to the back and forth nature of this procedure. The methodology section will therefore be revised to ensure fewer jumps in logic. As suggested, Section 2.6 on "Homogenisation" and section 2.7 on "Generation of daily and monthly series" shall be incorporated into a larger data processing section.
Parts of the "Source material description" could be shortened and included in the "Inventory". The "Inventory" section is actually part of the results and is an output of the study, as much as the rescued measurement series. We will more clearly explain this in the Introduction and Methods sections.

2. ***The study could be presented in a more constructive framework and its significance to the broader audience needs to be clearly emphasized.*** *For example, it would be more informative for the general audience, if the uncertainties and errors of the observations in different stations (section 2.5 on "Quality control") could be categorized by common characteristics, by regions, or even by specific years. In the demonstration of the two extreme weather cases, the reference to the CAP7 weather type lacks necessary explanation and justification.*

**Reply:** Agreed, the significance of the study to the broader audience and its contribution to different fields of research will be contextualized better in both the introduction and conclusion. Regarding the specific errors and uncertainties from each station may take up too much space; each type and occurrence of flagged values varies strongly from one station to another, for anyone wishing to know details, these flags are marked in the individual files in the data supplement. Section 2.5 "Quality control" seeks to highlight the main problems that were encountered by bringing up some examples, such as the wmo_time_consistency flags for Munich and Zagan, or the subdaily_repetitions for Moscow. We will evaluate how to better summarise uncertainties in this chapter.

A sentence will be added to clarify the reference to the CAP7 weather type classifications by Schwander et al. (2017).

3. *The authors should make a clearer distinguish between what's available to the public (i.e., the work already done by previous studies) on the Ephemerides and what's new from this reconstruction.*

**Reply:** We will better delineate how our work stands out from previous efforts that dealt with these observations. This is not a reconstruction but rather a rescue of observations contained in the *Ephemerides*, which includes: 1) the creation of an inventory that may serve future research as guide; 2) the conversion and correction of temperature and pressure measurements from 37 stations for use in modern climate research. It is the first time SMP data for one or more variables from all 37 stations is published. So far, studies have focused on the use of few series, and in most cases these have not been made publicly available.

4. *Please explain briefly why temperature and pressure, but not the other weather parameters, are specifically selected for the reconstruction. Could the quality of reconstruction on temperature and pressure be generalized to the other parameters? What implications does it have on the overall potential of the SMP dataset?*

**Reply:** The revised manuscript shall briefly justify the selection of the variables. The reason behind this choice lies in the fact that temperature and pressure are arguably two of the most fundamental parameters in any analysis of weather and climate. With regard to the *Ephemerides* this has its advantages, as thermometers and barometers were among the instruments issued by the SMP that were carefully calibrated and standardized to ensure more precise quantification. The same holds for the hygrometer, though it is not clear what the exact units being used are, so the data rescue for this parameter would require more effort. Precipitation is of course another important parameter; however, rain gauges in the SMP were designed for local construction and hence come in several different units, some of them based on specific regional subdivisions based on other weight units. This does not necessarily correspond to lower quality or reliability, but it does again mean the work involved to process them correctly would require tremendous effort. Even the descriptions of the state of the sky could be useful but would require time, care, and a different way of approaching data rescue, perhaps being processed as categorical variables or a set of indices.

This explanation could be added to the manuscript either in the introduction when framing the scope of the study or in an outlook within the conclusion

Overall, it is hard to generalize the quality of temperature and pressure to other variables. The potential here lies in the sheer quantity of recorded data; the fact that these observations were taken following a set of agreed-upon rules already makes them more reliable than many other series from this time. Within this project, almost all variables measured by the SMP have been digitized on

excel sheets and are waiting to be used, processed and tested for their usefulness in climate research.

5. *Based on Fig. 5 there are more than 25 stations available in 1785, so please explain why only observations of six stations are presented in Fig. 8. What about the rest of the stations? Did they also record the cold spell in March 1785?*

**Reply:** True, more than 25 stations gave data to the Society in 1785 and yes, they do show the cold spell (to a different extent depending on their location). The selection in Fig. 5 shows a number of stations in Central Europe, the region with most SMP stations but also the region that felt this cold spell more intensely. The selection was made to represent this Central area ranging from Mannheim to Budapest and from Padua to Zagan, with two additional stations in between. Furthermore, plotting all 25+ series on top of each other in Fig. 8 would have been too messy; whereas here one can still clearly distinguish which line belongs to which stations. Nevertheless, a figure with more station series could be added to the electronic supplement.

[Figure]

*Technical corrections:*

1. *Ln 63, "In a first part" sounds strange to me, please consider use the common notation "In the first part".*

**Reply:** Will be corrected.

2. *I don't understand the exact meaning of the sentence "Overall, the quality of the temperature and pressure series recovered in this study is relatively high – due in no small part to the standardized thermometers and barometers made available by the SMP" (Ln 184-185).*

**Reply:** The homogenization test showed that most of the pressure series and virtually all of the temperature series agreed with each other, and therefore needed no additional corrections. This is not surprising given that the thermometers and barometers given to the stations were standardized and calibrated with each other in such a way that they could be comparable (see Ln 88-94). "Relatively high" here means compared to other non-Palatina observations.
Instruments were in most cases quite good by the late 18th century, and problems with length units affected mainly precipitation (which the SMP did not solve). The most important thing that the SMP did is to spread best practices on how to use the instruments (e.g., isolating the outside thermometer from the wall with a wooden plate, measuring room temperature to later correct pressure readings). Providing standardized instruments with a common set of instructions to be gathered centrally was a good thing, though not because other instruments at the time were unreliable; it would just have been a lot more work to collect all information on each and every instrument and to make the necessary conversions/corrections (see Brugnara, 2015). Rescuing such

data for modern climate research requires a tremendous effort. These reasons make the SMP data even more valuable: the Society's insistence on precise standardized instruments and emphasis on coordinating observations to a common plan means that data rescue today can use these measurements with more confidence.

3. *Ln 287-288, Fig 5 does not give "which stations were more prolific", but how many stations. Please correct.*

**Reply:** True, it primarily shows how many stations contributed measurements, yet the last column "Period covered" shows the extent of each station's contribution, hence the use of the phrase "which stations were more prolific". Perhaps this can be expressed better.

4. *Ln 435-438, the whole sentence reads very confusing and needs clarification. What are "These differences" refer to precisely? What leads to "meaning that average March …"?*

**Reply:** Will be clarified and explained in a more correct manner. The differences refer to the Budapest station in Table 4: the Palatina data show an anomaly of -5.6°C for the period 1781-92, whereas EKF400v2 shows an anomaly of -3.6°C, meaning a difference of -2°C. This means that the SMP observations presented in this study consider March 1785 to be more anomalously cold than the reanalysis EKF400v2. This could be due to: 1) the winter 1788/9 in Budapest is warmer in EKF400v2 than in the SMP observations or 2) average March temperature for 1781-92 (used to calculate the anomaly) are lower in EKF400v2 than the SMP observations.

Brugnara, Y., et al.: A collection of sub-daily pressure and temperature observations for the early instrumental period with a focus on the "year without a summer" 1816, Clim. Past., 11, 1027-1047, https://doi.org/10.5194/cp-11-1027-2015, 2015.

---

## Author Response (AR1)

**Response to the review by Anonymous Referee #1 on the manuscript cp-2021-57 "Unlocking weather observations from the Societas Meteorologica Palatina (1781-1792)" by Duncan Pappert et al., including mention of relevant changes.**

We thank the reviewer for their effort in carefully reading and commenting on our manuscript. In the following, we reply to his comments point by point.

*Review: The database will undoubtedly be of importance to historical climatologists, and this manuscript could be useful to users of the database and to historical climatologists in general. However, as explained below, the article would be much more useful and impactful if the structure were revised for clarity. In particular, the authors should distinguish discussions and analysis of the original SMP observations from discussions and applications of the new database. Finally, the manuscript should address specific questions and concerns at the end of this review.*

*First, this structure can be confusing for the reader and makes it difficult to find and remember important content. Some topics are divided across multiple sections, while in other cases information has to be repeated. For example, sections 2.1 and 3.1 substantially overlap, and it's not clear why some information went in one section instead of the other. The fact that the original society reports contained much non-meteorological information and that this information was left out of the database seem like items that should have been discussed in "methods and data". On the other hand, much of the "methods and data" section consists of analysis or induction that I would normally expect in a "results" section (which is altogether missing from the article). These issues of clarity are problematic in an article such as this one, which is meant to serve as a guide to users.*

*Second, the structure does not allow the author to systematically address the project's context within, and contributions to, each field of research relevant to this study. These fields include:*

- *The history of meteorology and climatology;*
- *The study of early weather instruments and their measurements;*
- *Data rescue of early instrumental measurements;*
- *The creation of useful historical climatology databases; and*
- *High-resolution historical climate reconstruction.*

*I wouldn't expect detailed discussion of each of these. However, simply keeping them in mind and addressing each specifically in the introduction and conclusion would help frame the article and ensure it does not neglect important context or applications.*

*Third, the current structure blurs discussion and analysis of the methods and publications of the original SMP observation network and the usefulness of its measurements, on the one hand, with discussion and analysis of the methods and organization of the new database and its usefulness for climate reconstruction, on the other. Each of these is an interesting subject in its own right. As a researcher in historical climatology and potential user of the database, I would like to know (1) about the SMP, its background, and its activities; (2) about the new database and how it works; (3) what the database can tell us about the SMP, its coverage, and the quality of its observations; and (4) whether and how the database can help improve climate reconstructions. While that information is present in various places in the manuscript, it is currently all mixed together in ways that make it very hard to get at a good distinct explanation of each of these topics.*

*As referee, I would not want to dictate the structure of the article. There are various ways the authors could revise or amend the current structure to address the three major issues outlined above. However, I suggest the following possible changes:*

- − *The introduction could address the project's context within, and contributions to, the relevant research fields, as described above. The background and history of the SMP would fit better in the section on sources and methods.*

- − *The section on sources and methods could separate (1) the structure, methods, and publications of the SMP from (2) the structure of the database, its methods of compiling SMP data, the methods used to analyze the SMP network with the database, methods of correcting and homogenizing SMP measurements as they are put into the database, and how the database could be compared to other data or studies to examine its value for reconstruction.*

- − *There could be a "results" section presenting (1) the results of the analyses of SMP data for coverage, resolution, and consistency (currently in section 3.1); (2) results of analyses of tests for homogeneity and breakpoints (currently in section 2); (3) results of comparisons with other data or reconstructions (currently in section 4).*

- − *A "discussion" section could include much of the discussion in section 3.2 and the conclusion.*

- − *The conclusion could recap the article and address the project's contribution to the relevant fields of study presented in the introduction.*

**Reply:** Following almost exactly the suggestions outlined above, the structure of the manuscript underwent a major revision. The introduction and conclusion now better frame the context of the study in relation to its significance for various fields of research. The methods section is now divided into two main parts, the first dealing with the background of the network and the source materials, the second with the data rescue procedure of the measurements. The results section now incorporates the outcome from the data rescue with relation to data coverage and quality; additionally, we present the two case studies of extreme weather as a demonstration of the data's potential usefulness for further studies. The discussion includes much of what was previously Sect. 3.2 (some of it also moved to now Sect. 2.1): this part now seeks to explain some of the patterns in the inventory and interprets the results and implications of the data rescue, as well as discusses the potential of the new dataset.

**Review:** *Is the database compatible with those of other early instrumental data rescue projects such as ACRE? Are the results being incorporated into ACRE or another such project?*

**Reply:** Two of the co-authors were directly involved in the establishment of the Copernicus Data Rescue Service, which is tightly related with ACRE, and collaborate regularly with ACRE members. The data format complies with the best practices recommended by Copernicus (https://climate.copernicus.eu/sites/default/files/2021-05/C3S_DC3S311a_Lot1.3.4.2_2020_BestPracticeGuidelines_Part2.pdf). The data will be eventually submitted to the Copernicus database (GLAMOD).

**Review:** *Have any of the original instruments from the SMP (or other examples of the same manufacture) survived? If so, have they been analyzed for any particular errors or biases, particularly those that would require non-linear corrections at high or low temperatures (in addition to the data processing already discussed in section 2.3)?*

**Reply:** We do not know of any instrument that survived, although certainly some must have. We visited the astronomical observatories in Geneva and Bologna (i.e., the two Palatina stations closest to Bern) but did not find any instrument from the Palatina in their collections. In any case, the instruments reflected the standard for the time and were already rather accurate (see, e.g., the literature from the IMPROVE project – Camuffo and Jones, 2002). In fact, reliable international calibration standards were already in place (Cavendish, 1777). The main instrumental error source that we are aware of is the contraction of the glass, and that is accounted for in our corrections. Another known problem of the Palatina thermometer is that it was only graduated until -16°Ré, which makes lower temperatures somewhat less accurate (we will add this information to the manuscript).
Larger, non-linear errors are probably related to the exposure of the thermometer, but they are hard to estimate and out of the scope of this paper, as they affect pretty much any temperature measurement up until at least the early 20th century.

*Review: The case study in section 4.1 compares information in the new database with the reconstructions of Kington 1988. However, as described in the introduction (line 55-56), Kington 1988's reconstruction is also based on SMP measurements. Therefore, the comparison does not justify the conclusion that "This brief example demonstrates the reliability of the processed pressure data as well as its usefulness when paired with documentary sources" (lines 426-427). For tests of reliability, the processed SMP data should be compared with other early instrumental series or high-resolution reconstructions based on reliable phenological observations. In fact, if there were differences between Kington 1988 and the new database, then that would actually underline the importance of the correction and homogenization applied to the SMP data. Instead, this example seems to illustrate relative ease of producing and visualizing data with the new database (as opposed to the laborious hand-drawn maps of Kington).*
*Similarly, in section 4.2, can the authors confirm that the EKF400v2 reanalysis (line 431) did not itself use SMP measurements?*

**Reply:** Yes, 'reliability' may be the wrong choice of words as it implies a validation against high-quality independent observations. This is not strictly the case here as both Kington's hand-drawn maps and EKF400 used SMP data (although in the latter case, only a handful of series were used). The examples in the study are meant to showcase the usefulness of the recovered series, their agreement with each other, and how they fit with and complement existing work done on this period. Combined with information from historical databases, the potential of these measurements could be further uncovered.
As the main added value of this database in comparison with existing products is its daily resolution, there are no independent, high-quality products with that resolution that we can validate against.

*Review: The manuscript is generally well written but contains occasional minor issues of English usage (for example, in line 30 "spurn" should be "spur").*

**Reply:** This was corrected and the manuscript was scanned for similar mistakes.

Camuffo D., Jones P. (eds), 2002, Improved Understanding of Past Climatic Variability from Early Daily European Instrumental Sources. Springer, Dordrecht. https://doi.org/10.1007/978-94-010-0371-1_1.

Cavendish, H., Heberden, Alex-Aubert, Deluc, Maskelyne, Horsley, Planta, 1777, The report of the Committee of the Royal Society to consider of the Best Method of Adjusting the Fixed Points of Thermometers; and the Precautions Necessary to be Used in Making Experiments with these Instruments, Philosophical Transactions 67, 816–857.

**Response to the review by Anonymous Referee #2 on the manuscript cp-2021-57 "Unlocking weather observations from the Societas Meteorologica Palatina (1781-1792)" by Duncan Pappert et al., including mention of relevant changes.**

We thank the reviewer for their effort in carefully reading and commenting on our manuscript. In the following, we reply to his comments point by point.

*Review: This study reported the digitizing and compilation of the Societas Meteorologica Palatina (SMP) weather observation, a network of 37 stations across Europe plus a couple in North America and Greenland coving the decade of 1781 and 1792. The quality of the SMP temperature and pressure observation data is then evaluated by using the C3S Quality Control software to identify outliers and visual inspection. The potential of the SMP reconstruction for climate research is demonstrated by two extreme weather events of the reconstructed period. The reconstruction is rigorously done and described in great detail. However, the manuscript, in its current layout, requires major revision before it could be considered for publication in this journal.*

*Specific comments:*

1. ***The structure of the manuscript is not logically sound.*** *For example, section 2.1 on "Source material description" and section 3 on "Inventory" could be combined and shortened by removing the repeating or loosely relevant information. Section 2.6 on "Homogenisation" and section 2.7 on "Generation of daily and monthly series" should be parts of the "Data processing (section 2.3)" work. Section 2.7 on "Generation of daily and monthly series" should also be placed before section 2.5 on "Quality control" and section 2.6, since a number of discussions in quality control and homogenization refer to the monthly data.*

**Reply:** The structure of the manuscript was entirely revised, following specific suggestions made by Anonymous Reviewer #1. The old Inventory (Sect. 3) was changed, much of its content went to now Sect. 2.1 on the background of the SMP and description of the publications, while the rest went towards forming a Discussion section (Sect. 4). The section on homogenization and generation of series went into the Data processing seciont (now Sect. 2.2.2). The outcome of the quality control and homogenization tests went into part of the results, (now) Sect. 3.2.

2. ***The study could be presented in a more constructive framework and its significance to the broader audience needs to be clearly emphasized.*** *For example, it would be more informative for the general audience, if the uncertainties and errors of the observations in different stations (section 2.5 on "Quality control") could be categorized by common*

*characteristics, by regions, or even by specific years. In the demonstration of the two extreme weather cases, the reference to the CAP7 weather type lacks necessary explanation and justification.*

**Reply:** Along with the restructuring of the manuscript, we largely rewrote the introduction and conclusion sections to better place the study and its significance to specific research fields. We left the content of the quality control (now) Sect 3.2 as it was, which placed more emphasis on the types of errors themselves, to see from what type of error did the SMP data suffer mostly. Each SEF file in the supplement that underwent QC lists if a given value failed a test. We added a sentence in (now) Sect. 2.2.3. to clarify the later reference to the CAP7 weather type classifications by Schwander et al. (2017) in the case studies.

3. *The authors should make a clearer distinguish between what's available to the public (i.e., the work already done by previous studies) on the Ephemerides and what's new from this reconstruction.*

**Reply:** We will better delineate how our work stands out from previous efforts that dealt with these observations. This is not a reconstruction but rather a rescue of observations contained in the *Ephemerides*, which includes: 1) the creation of an inventory that may serve future research as guide; 2) the conversion and correction of temperature and pressure measurements from 37 stations for use in modern climate research. It is the first time SMP data for one or more variables from all 37 stations is published. So far, studies have focused on the use of few series, and in most cases these have not always been made publicly available.

4. *Please explain briefly why temperature and pressure, but not the other weather parameters, are specifically selected for the reconstruction. Could the quality of reconstruction on temperature and pressure be generalized to the other parameters? What implications does it have on the overall potential of the SMP dataset?*

**Reply:** We justified the selection of variables in (now) Sect. 2.2.2.
Here we give a more extensive answer. This research paper came out of the MSc Thesis of D. Pappert, in which he chose to study these two variables with which he subsequently performed some climatological analysis. Part of the reason behind this choice lies in the fact that temperature and pressure are arguably two of the most fundamental parameters in any analysis of weather and climate. With regard to the *Ephemerides* this has its advantages, as thermometers and barometers were among the instruments issued by the SMP that were carefully calibrated and standardized to ensure more precise quantification. The same holds for the hygrometer, though it is not clear what the exact units being used are, so the data rescue for this parameter would require more effort. Precipitation is of course another important parameter; however, rain gauges in the SMP were designed for local construction and hence come in several different units, some of them based on specific regional subdivisions based on other weight units. This does not necessarily correspond to lower quality or reliability, but it does again mean the work involved to process them correctly would require tremendous effort. Even the descriptions of the state of the sky could be useful but would require time, care, and a different way of approaching data rescue, perhaps being processed as categorical variables or a set of indices.

Overall, it is hard to generalize the quality of temperature and pressure to other variables. The potential here lies in the sheer quantity of recorded data; the fact that these observations were taken following a set of agreed-upon rules already makes them more reliable than many other series from this time. Within this project, almost all variables measured by the SMP have been digitized on excel sheets and are waiting to be used, processed and tested for their usefulness in climate research.

5. *Based on Fig. 5 there are more than 25 stations available in 1785, so please explain why only observations of six stations are presented in Fig. 8. What about the rest of the stations? Did they also record the cold spell in March 1785?*

**Reply:** True, more than 25 stations gave data to the Society in 1785 and yes, they do show the cold spell (to a different extent depending on their location). The selection in Fig. 5 (now 8) shows a number of stations in Central Europe, the region with most SMP stations but also the region that felt this cold spell more intensely. The selection was made to represent this central area ranging from Mannheim to Budapest and from Padua to Zagan, with two additional stations in between.
Furthermore, plotting all 25+ series on top of each other in this figure would have been too messy; it is important to us not only to show agreement between the stations, but also to be able to distinguish which line/color belongs to which stations. We have added another version of this figure in the supplement with several more series.

[Figure]

*Technical corrections:*

1. *Ln 63, "In a first part" sounds strange to me, please consider use the common notation "In the first part".*

**Reply:** Has been changed.

2. *I don't understand the exact meaning of the sentence "Over all, the quality of the temperature and pressure series recovered in this study is relatively high – due in no small part to the standardized thermometers and barometers made available by the SMP" (Ln 184-185).*

**Reply:** The homogenization test showed that most of the pressure series and virtually all of the temperature series agreed with each other, and therefore needed no additional corrections. This is not surprising given that the thermometers and barometers given to the stations were standardized and calibrated with each other in such a way that they could be comparable (see Ln 88-94). "Relatively high" here means compared to other non-Palatina observations.
Instruments were in most cases quite reliable by the late 18th century, and problems with length units affected mainly precipitation (which the SMP did not solve). The most important thing that the SMP did is to spread best practices on how to use the instruments (e.g., isolating the outside thermometer from the wall with a wooden plate, measuring room temperature to later correct pressure readings). Providing standardized instruments with a common set of instructions to be gathered centrally was a good thing, though not because other instruments at the time were unreliable; it would just have been a lot more work to collect all information on each and every instrument and to make the necessary conversions/corrections (see Brugnara, 2015). Rescuing such data for modern climate research requires a tremendous effort. These reasons make the SMP data even more valuable: the Society's insistence on precise standardized instruments and emphasis on

coordinating observations to a common plan means that data rescue today can use these measurements with more confidence.

   3. *Ln 287-288, Fig 5 does not give "which stations were more prolific", but how many stations. Please correct.*

**Reply:** This has been corrected.

   4. *Ln 435-438, the whole sentence reads very confusing and needs clarification. What are "These differences" refer to precisely? What leads to "meaning that average March …"?*

**Reply:** We have clarified this sentence in the revised manuscript.
The differences refer to the Budapest station in Table 4: the Palatina data show an anomaly of -5.6°C for the period 1781-92, whereas EKF400v2 shows an anomaly of -3.6°C, meaning a difference of -2°C. This means that the SMP observations presented in this study consider March 1785 to be more anomalously cold than the reanalysis EKF400v2. This could be due to: 1) the winter 1788/9 in Budapest is warmer in EKF400v2 than in the SMP observations or 2) average March temperature for 1781-92 (used to calculate the anomaly) are lower in EKF400v2 than the SMP observations.

Brugnara, Y., et al.: A collection of sub-daily pressure and temperature observations for the early instrumental period with a focus on the "year without a summer" 1816, Clim. Past., 11, 1027-1047, https://doi.org/10.5194/cp-11-1027-2015, 2015.